# The importance of an informed choice of $CO_2$-equivalence metrics for contrail avoidance

Audran Borella[1], Olivier Boucher[1], Keith P. Shine[2], Marc Stettler[3], Katsumasa Tanaka[4,5], Roger Teoh[3] and Nicolas Bellouin[1,2]

[1]Institut Pierre-Simon Laplace, Sorbonne Université / CNRS, Paris, France
[2]Department of Meteorology, University of Reading, Reading, UK
[3]Centre for Transport Studies, Department of Civil and Environmental Engineering, Imperial College London, London, UK
[4]Laboratoire des Sciences du Climat et de l'Environnement (LSCE), IPSL, CEA/CNRS/UVSQ, Université Paris-Saclay, Gif−sur−Yvette, France
[5]Earth System Division, National Institute for Environmental Studies (NIES), Tsukuba, Japan

*Correspondence to*: Audran Borella (audran.borella@ipsl.fr)

**Abstract.** One of the proposed ways to reduce the climate impact of civil aviation is rerouting aircraft to minimise the formation of warming contrails. As this strategy may increase fuel consumption, it would only be beneficial if the climate impact reduction from the avoided contrails exceeds the negative impact of any additional carbon dioxide ($CO_2$) emitted by the rerouted flight. In this study, we calculate the surface temperature response of almost half-a-million flights that crossed the North Atlantic sector in 2019 and compare it to the temperature response of hypothetical rerouted flights. The climate impacts of contrails and $CO_2$ are assessed through the perspective of $CO_2$-equivalence metrics, represented here as nine combinations of different definitions and time horizons. We estimate that the total emitted $CO_2$ and the persistent contrails formed will have warmed the climate by 17.2 µK in 2039, 13.7 µK in 2069, and 14.1 µK in 2119. Under an idealised scenario where 1% additional carbon dioxide is enough to reroute all contrail-forming flights and avoid contrail formation completely, total warming would decrease by 4.9 (−28%), 2.6 (−19%), and 1.9 (−13%) µK in 2039, 2069, and 2119, respectively. In most rerouting cases, the results based on the nine different $CO_2$-equivalence metrics agree that rerouting leads to a climate benefit, assuming that contrails are avoided as predicted. But the size of that benefit is very dependent on the choice of $CO_2$-equivalence metrics, contrail efficacy and $CO_2$ penalty. Sources of uncertainty not considered here could also heavily influence the perceived benefit. In about 10% of rerouting cases, the climate damage resulting from contrail avoidance indicated by $CO_2$-equivalence metrics integrated over a 100-year time horizon is not predicted by metrics integrated over a 20-year time horizon. This study highlights, using North Atlantic flights as a case study, the implications of the choice of $CO_2$-equivalence metrics for contrail avoidance, but the choice of metric implies a focus on a specific climate objective, which is ultimately a political decision.

## 1 Introduction

Civil aviation mostly contributes to climate change through the in-flight emissions of commercial aircraft. These emissions include carbon dioxide ($CO_2$): the global aviation fleet represented about 2.4% of the global annual fossil-fuel $CO_2$ emissions in 2018 (ICCT, 2018). However, kerosene combustion also emits non-$CO_2$ species such as nitrogen oxides (NOx), water vapour, and soot particles. These emissions exert the so-called non-$CO_2$ effects of aviation (EASA, 2020). The $CO_2$ and non-$CO_2$ climate effects have usually been compared in terms of radiative forcing (e.g., Ramaswamy et al., 2018) or other dedicated metrics (e.g., Fuglestvedt et al., 2010). According to current knowledge, non-$CO_2$ effects could exert two thirds of the total effective radiative forcing due to civil aviation since its inception in 1940 (Lee et al., 2021). Uncertainties in the estimates of non-$CO_2$ effects are large and the subject of active research.

Contrails (condensation trails) form when atmospheric conditions allow water vapour to condense onto aerosols emitted by the incomplete combustion of kerosene impurities in the aircraft engines (Schumann et al., 2012; Kärcher, 2018). Contrails that form in ice supersaturated regions (ISSRs) of the upper atmosphere persist to form contrail cirrus clouds. These clouds reflect a fraction of incoming sunlight back to space, cooling the climate system, but they also absorb and re-emit infrared radiation at colder temperatures than the underlying surface and atmosphere, leading to a warming. The overall impact depends on the contrail properties, its ambient environment, and the time of day. Some persistent contrails cool the Earth, but most have a warming effect (Wilhelm et al., 2021; Wang et al., 2023). Taken together, contrail cirrus are currently thought to be the largest contributor of climate forcing by aviation (Figure 6.16 of Szopa et al. 2021) and hence are the focus of this study. A more comprehensive discussion would need to assess the impact of other non-$CO_2$ effects, such as NOx and aerosols. There is an interdependency between them: for example, aerosols play a role in contrail formation but also exert their own radiative forcing, directly by interacting with radiation or via their interactions with clouds, but the magnitude and size of this forcing is highly uncertain (Lee et al., 2021).

On average, 13% of flight time occurs in ISSRs (Gierens et al., 1999), and studies suggest that only a minority of flights in a given airspace are responsible for most of the higher-impact persistent contrails (Teoh et al., 2020; 2022; 2023). It is therefore tempting to avoid flight trajectories or altitudes that encounter ISSRs to avoid the formation of persistent contrails. This strategy assumes that the location and properties of ISSRs can be accurately predicted on both the original and rerouted trajectories, and it also assumes that other considerations, including impacts on air traffic management, safety and flight duration, make a reroute practical. In addition, the rerouted aircraft could deviate from its optimal trajectory in terms of fuel consumption and therefore $CO_2$ emissions, thus requiring methods to quantify the trade-off between $CO_2$ and non-$CO_2$ climate impacts.

The radiative and climate impacts of $CO_2$ and non-$CO_2$ effects are usually compared using a $CO_2$-equivalence metric as a multiplicative factor that operates on the $CO_2$-only effect (Fuglestvedt et al., 2010). For example, a factor of 2 indicates a net $CO_2$ and non-$CO_2$ effect that is twice as large as that of the $CO_2$ effect alone, while a factor of 1 indicates that the net non-$CO_2$ effects are zero. The multiplicative factor can thus translate emissions of aviation into $CO_2$-equivalent emissions.

It is now well accepted that the choice of $CO_2$-equivalence metric depends on the perspective and objectives of climate policies (Fuglestvedt et al., 2010; Tanaka et al., 2010; Forster et al., 2021; Megill et al., 2024). The UNFCCC (United Nations Framework Convention on Climate Change) decided in 1995 for its Kyoto protocol to compute $CO_2$-equivalent emissions of greenhouse gas emissions using a metric definition based on the integration of radiative forcing, called the Global Warming Potential (GWP), over a time horizon of 100 years. That decision was reconfirmed in 2018 (UNFCCC, 1995, 2019). The initial choice was informed by the First Assessment Report of the Intergovernmental Panel on Climate Change (IPCC), which assessed GWP for 20-, 100-, and 500-year time horizons in its Table 2.8 (Shine et al., 1990). The report noted that these "three different time horizons are presented as candidates for discussion and should not be considered as having any special significance" (Shine, 2009). UNFCCC therefore had a limited choice of $CO_2$-equivalence metric, since GWP was the only metric assessed by the IPCC at the time. It is believed that the 100-year time horizon may have been chosen as a middle-ground between shorter and longer options. The EU Emission Trading System followed in 2003 the UNFCCC choice of GWP (European Commission, 2003, Directive 2003/87/EC). This decision does not mandate a particular time horizon, but 100 years is likely to remain the most widely used. In its most recent assessment (AR6), the IPCC refrained from recommending an emission metric (Box 7.3 of Forster et al. 2021), noting that the choice "depends on the purposes for which [...] forcing agents are being compared". Although the UNFCCC choice of GWP100 was based on the limited research available in the mid-1990s, it is becoming clearer that GWP100 is relatively close to an economically optimal metric at least currently for achieving the long-term temperature goals of the Paris Agreement (see Cross Chapter 2 of Dhakal et al., 2022; Tanaka et al., 2021).

In the case of aviation, Grewe et al. (2014) studied the impact of metric choice on estimating the short, medium, and long-term climate impacts of re-routing strategies, and found that some combinations of metrics and time horizon put a greater value on reducing the radiative forcing of short-lived components like contrails, while other combinations put a greater value on reducing the forcing due to $CO_2$. Aviation is not the only sector where non-$CO_2$ emissions are a significant fraction of the $CO_2$-equivalent emissions. Most notable are methane emissions from the agricultural sector for which the $CO_2$-equivalence is also heavily dependent on the choice of metric and time horizon, and the debate and lack of scientific consensus on that choice continues, as noted in the latest report of the Food and Agriculture Organization (FAO) on the topic (FAO, 2023).

The present study revisits the question of the choice of a $CO_2$-equivalence metric definition and time horizon in the context of contrail avoidance, using the actual 2019 North Atlantic air traffic to quantify the climate outcome of that choice. It includes a sensitivity analysis of the results to the amount of additional $CO_2$ needed to reroute a flight, to the capability of the rerouting to fully avoid the formation of a persistent contrail, and to the efficacy of contrail radiative forcing to induce surface temperature changes. Section 2 details the metrics used to quantify the impact of contrails, and the $CO_2$-equivalence metric definition that allows comparison of $CO_2$ and contrails climate impacts. Section 3 describes the air traffic dataset used in this study and the contrail and Earth system models used to calculate the surface temperature change exerted by a flight based on its $CO_2$ emissions and contrail radiative forcing. Section 4 discusses how the different time scales of the climate perturbations of $CO_2$ and contrails affect the results based on different $CO_2$-equivalence metrics, focusing on nine metrics,

which are the combinations of three different metric definitions (global warming potential, global temperature change potential, and average temperature response) with three alternative time horizons (20, 50, and 100 years). Section 5 uses case studies to illustrate how the climate outcome of re-routing decisions depends on the combination of metric definition and time horizon. Section 6 assesses the climate implications of re-routing strategies applied to contrail-forming aircraft traffic over the North Atlantic for the year 2019, with simplified assumptions on the additional $CO_2$ emitted to avoid contrail formation. Sections 7 and 8 investigate the sensitivity of the results to more realistic rerouting and to contrail efficacy, respectively. Section 9 concludes the study, highlighting the implications of radiative forcing uncertainties.

## 2. CO₂-equivalence metrics

It is impractical to run climate models to quantify the different contributions to temperature change for individual activities, or individual flights in our case, because the climate impact of such a small emission pulse would be lost in the internal variability of the Earth system, requiring too many climate model simulations to identify it robustly. Instead, different $CO_2$-equivalence metrics (also called "climate metrics", "emission metrics" or simply "metrics" in the literature) have been developed to compare $CO_2$ and non-$CO_2$ effects (Fuglestvedt et al., 2010). Such a comparison can be done in terms of the time evolution of the radiative forcing $\Delta F$ exerted by the constituent, or of the temperature change $\Delta T$ that follows from $\Delta F$. The Absolute Global Warming Potential (AGWP) measures the cumulative radiative forcing exerted by a pulse emission at $t_0$ of the constituent of interest over a given time horizon $H$ (Fuglestvedt et al., 2010):

$$\text{AGWP}_\text{H} = \int_{t_0}^{t_0+H} \Delta F(t) dt \tag{1}$$

Relating the $\text{AGWP}_\text{H}$ of contrails to a pulse emission is not straightforward, since contrails are not emitted. Instead, contrails form owing to the release of water vapour and aerosols from fuel combustion, and one estimates the $\text{AGWP}_\text{H}$ of a contrail formed from the energy it adds to or removes from the Earth system during its lifetime. $\text{AGWP}_\text{H}$ is a time-integrated metric, and because it is based on radiative forcing, it is not an explicit measure of the climate response (Fuglestvedt et al., 2010). To consider the climate response, the Absolute Global Temperature Change Potential (AGTP; Shine et al., 2005) measures the change in global surface temperature at time horizon $H$ after a pulse emission at $t_0$:

$$\text{AGTP}_\text{H} = \Delta T(t_0 + H) \tag{2}$$

Unlike $\text{AGWP}_\text{H}$, $\text{AGTP}_\text{H}$ is an endpoint metric. Corresponding integrated metrics have been proposed that measure global surface temperature change averaged over $H$: the integrated GTP (Peters et al., 2011), mean GTP (Gillett and Matthews, 2010), or Average Temperature Response (ATR; Grewe et al. 2021). The latter has been introduced in a slightly different form by Dallara et al. (2011), then used regularly in aviation-climate studies following the publication of Grewe et al. (2014), so will be considered here. $\text{ATR}_\text{H}$ is the average change in temperature between the time of an emission at $t_0$ and the end of the time horizon $t_0 + H$:

$$\text{ATR}_\text{H} = \frac{1}{H} \int_{t_0}^{t_0+H} \Delta T(t) dt \tag{3}$$

Hereafter, we denote the unit of ATR with brackets <K>, to distinguish an average temperature over a time period from an endpoint temperature change.

Combined pulse/sustained metrics, like the GWP* (Allen et al. 2018) and CGTP (Collins et al. 2020), have been proposed to ensure equivalency in the temperature response of short-lived and long-lived climate forcers (Allen et al., 2022). Those metrics are not considered here because they are most suited for quantifying $CO_2$/non-$CO_2$ trade-off for future emission scenarios rather than for quantifying potential reductions in climate impact for individual flights, which is the topic of this study.

Note that, as detailed in the next section, we calculate the time-dependent temperature response, then apply the perspective of $CO_2$-equivalence metrics to interpret the results, without relying on previously published values of $CO_2$-equivalence metrics (see Cross Chapter 2 of Dhakal et al., 2022).

## 3. Methods

This study considers flight trajectories, their associated fuel consumption and $CO_2$ emissions, and, when relevant, the
persistent contrails formed. Each specific persistent contrail is described using its energy forcing $EF_\text{contrail}$ (Teoh et al., 2020), in joules, which quantifies the total energy added to or lost by the Earth system by the contrail during its lifetime while accounting for the length of the contrail $L$, its width $W$, and lifetime $\tau$:

$$EF_\text{contrail} = \int_0^\tau \Delta F_\text{contrail}(t) \times W(t) \times L(t) \times dt, \tag{4}$$

where $\Delta F_\text{contrail}$ is the instantaneous contrail RF. $EF_\text{contrail}$ is almost equal to $\text{AGWP}_\text{H}$ (but in a different unit) for time
horizons greater than 1 year (see Supplementary Data of Irvine et al., 2014). Such time horizons are much longer than the lifetime of a contrail, therefore $\text{AGWP}_\text{H}$ integrates the whole contrail radiative forcing. However, the equality is not exact because of the existence of slow carbon-cycle feedbacks within the Earth system, and $\text{AGWP}_\text{H}$ is about 7, 10 and 13% greater than $EF_\text{contrail}$ for a time horizon of 20, 50 and 100 years, respectively.

This study considers all segments of flights that operated over the North Atlantic in the Shanwick and Gander Oceanic
Control Areas, which are approximately bounded by longitudes 50°W and 10°W and latitude 40°N and 75°N. We use the data from the year 2019, before the disruption caused to intercontinental routes by the Covid-19 pandemic. That represents a total of 477,923 flights.

The dataset provides fuel consumption and $EF_\text{contrail}$, which were calculated by Teoh et al. (2022) using: (i) actual flight trajectories provided by the UK National Air Traffic Services (NATS); (ii) the European Centre for Medium Range Weather
Forecasts (ECMWF) ERA5 high-resolution realisation (HRES) reanalysis, which, for the purposes of this study, is presumed to be completely accurate; and (iii) the contrail cirrus prediction model CoCiP (Schumann et al., 2012). CoCiP tests the

meteorological conditions over flight segments and if they are favourable to contrail formation, a contrail segment is formed. CoCiP then advects the contrail in a Lagrangian framework to simulate coverage and radiative properties over the contrail lifetime. In this study, $EF_{contrail}$ is expressed in terms of energy forcing per flown distance, dividing total energy forcing by total flown distance for each flight.

According to those calculations, 260,854 flights (55%) formed contrails and 208,965 flights (80% of contrail-forming flights) formed contrails that exerted a positive contrail energy forcing. Contrail energy forcing per flown distance ranges from $-8.7\times10^{11}$ to $5.0\times10^{12}$ J km$^{-1}$, with a mean of $9.6\times10^{10}$ J km$^{-1}$ and a median of $2.9\times10^{10}$ J km$^{-1}$. The 10$^{th}$ and 90$^{th}$ percentiles are $-6.0\times10^{9}$ and $2.8\times10^{11}$ J km$^{-1}$, respectively. The mean energy forcing per unit flown distance of warming contrails is $1.3\times10^{10}$ J km$^{-1}$, and $-2.3\times10^{10}$ J km$^{-1}$ for cooling contrails. Although cooling contrails are associated with 20% of contrail-forming flights, their relatively weak energy forcing leads to removing a total energy of $2.8\times10^{18}$ J from the climate system, while warming contrails add $6.6\times10^{19}$ J to it, i.e. 23 times more in absolute terms. It should be noted that the uncertainties associated with those energy forcing estimates are large. Uncertainties were estimated by Teoh et al. (2022) but are left out of the present analysis.

Contrail energy forcing and emitted $CO_2$ are enough to calculate the radiative forcing of each flight but are not directly comparable quantities. So this study uses the OSCAR reduced-complexity Earth system model in its version 3.1.1 (Gasser et al., 2017a; 2020) to compute for each individual flight, with a 1-year timestep, the time evolution of the globally averaged radiative forcing and the globally averaged surface temperature change that occurs in response to that forcing. For $CO_2$, OSCAR computes the time evolution of stratospheric-adjusted radiative forcing (Hansen et al., 2005; Forster et al., 2007) from the provided flight $CO_2$ emissions using a multi-region box model with the empirical formula for $CO_2$ radiative forcing from Myhre et al. (2013). For contrails, energy forcing, in joules, is added to the climate system uniformly over 1 year, which is the time step of the model. That is done without geographical considerations. The use of instantaneous contrail RF (corresponding to EF) for contrails and the use of stratospheric-adjusted RF for $CO_2$ is not inconsistent, as instantaneous and stratospheric-adjusted RF do not differ significantly for contrails (Dietmüller et al., 2016). OSCAR calculates the climate response to radiative forcing by using an impulse response function. The carbon cycle response to a pulse $CO_2$ emission depends on the background concentration of $CO_2$. That background concentration is simulated using the emission scenario SSP4-3.4 in our experiment, which is characterised by a low atmospheric $CO_2$ concentration in comparison to the other pathways. However, conclusions are qualitatively independent on the chosen scenario.

Uncertainties in radiative forcing and surface temperature change calculated by OSCAR are assessed using a Monte Carlo approach based on 2000 simulations with different model parameters. We disregard 274 simulations that did not converge to a solution, which can arise with the Monte Carlo approach, and use the remaining 1726 simulations to provide the best estimate and standard deviation.

CoCiP and OSCAR do not simulate the tropospheric and stratospheric adjustments triggered by the contrail climate forcing. A contrail efficacy factor is used here for contrails and contrail cirrus to account for those adjustments: the contrail radiative forcing calculated by Teoh et al. (2022) is thus multiplied by contrail efficacy before being used in OSCAR. Contrail

efficacy is divided in two parts: the ERF-to-RF ratio and theERF-based efficacy (Hansen et al., 2005; Ponater et al, 2021; Bickel, 2023). The ERF-to-RF ratio quantifies the atmospheric adjustments that follow contrail formation, i.e., changes in stratospheric and tropospheric temperature, humidity, or cloudiness that happen without large-scale changes in surface temperature (Sherwood et al. 2015). The ERF-based efficacy quantifies the ability of a unit of contrail ERF to change surface temperatures compared to a unit of $CO_2$ ERF, noting that there is a small dependence on the efficacy of the assumed size of the $CO_2$ perturbation (Hansen et al., 2005). It can differ from 1 because different ERF patterns trigger different surface temperature responses (e.g., Hansen et al., 2005; Shindell and Faluvegi, 2009; Richardson et al., 2019). Total contrail efficacy is the product of these two efficacies. It is set to 0.37 in this study, which is the average of the three available estimates in the literature, specifically 0.59 (Ponater et al., 2005), 0.31 (Rap et al., 2010) and 0.21 (Bickel, 2023). The same factor is assumed here to apply to all contrails, but it is likely that contrail efficacy varies depending on the location and time of a flight. Nevertheless, this assumption is adopted here in the context of this study, where results are considered an average over a population of North Atlantic flights. Contrail RF is multiplied with total contrail efficacy before running OSCAR, and is taken into account in all the results in this study, in particular when using the AGWP (Fuglestvedt et al., 2003, their Eq. 7). The sensitivity of the climate outcome of contrail avoidance to the uncertainty in total contrail efficacy is investigated in Section 5.3.

## 4. $CO_2$ and contrail time scales

The climate perturbations of $CO_2$ and contrails are associated with very different time scales. These time scales are illustrated in an idealised but representative way by Figure 1, which shows the radiative forcing and surface temperature response caused by a typical contrail-forming flight flying 2450 km in the Shanwick and Gander sectors of the North Atlantic. This typical segment of flight burns 17 tons of kerosene and emits a total of 53 tons of $CO_2$ into the atmosphere. We assume in addition that the flight is responsible for forming a contrail cirrus with an energy per flown distance of $2.9 \times 10^{10}$ J km$^{-1}$, which is the median value for contrail-forming flights in the dataset.

The emission of $CO_2$ molecules from kerosene combustion is instantaneous, while the formation of the ice crystals that form persistent contrails and contrail cirrus takes a few seconds (Kärcher, 2018). Both the $CO_2$ molecules or ice crystals will exert a radiative forcing for as long as their respective atmospheric concentrations remain perturbed. The radiative forcing exerted by the contrail ice crystals, which is assumed in Figure 1 to be positive, lasts up to 10-15 hours (Kärcher, 2018), except for very small carbon cycle adjustments that develop over a few decades (Gasser et al., 2017b). In stark contrast, the emission of fossil $CO_2$ causes an initial increase in the $CO_2$ concentration in the atmosphere, and about 30% of this increase persists after 100 years, and even about 20% persists after 1000 years, thus exerting a radiative forcing long in the future (Archer et al., 2009; Joos et al., 2013). Note however that although the radiative forcing of contrail operates on a time scale orders of magnitude shorter than that of $CO_2$, it is also orders of magnitude stronger.

The stark differences in $CO_2$ and contrail radiative forcing time scales do not fully propagate to the temperature response (bottom row of Figure 1) because the ocean absorbs the energy perturbation resulting from the radiative forcing over the time the forcing is exerted, increasing its heat content, before returning that heat to the atmosphere over several decades (Stjern et al., 2023). $CO_2$ is associated with both short and long timescales because its radiative forcing is exerted over many years, effectively providing a slowly decreasing source of energy for the ocean to absorb and then release. The warming of contrails is felt by the climate system through the ocean response over several decades despite contrail cirrus only lasting for several hours. This long-term response is further amplified by so-called carbon cycle feedback, which causes the small increase in contrail radiative forcing that can be seen in Figure 1 a few years after the contrail has dissipated. That feedback occurs because warming leads to a release of $CO_2$, primarily through increased decomposition of soil organic matter, which in turn leads to more warming.

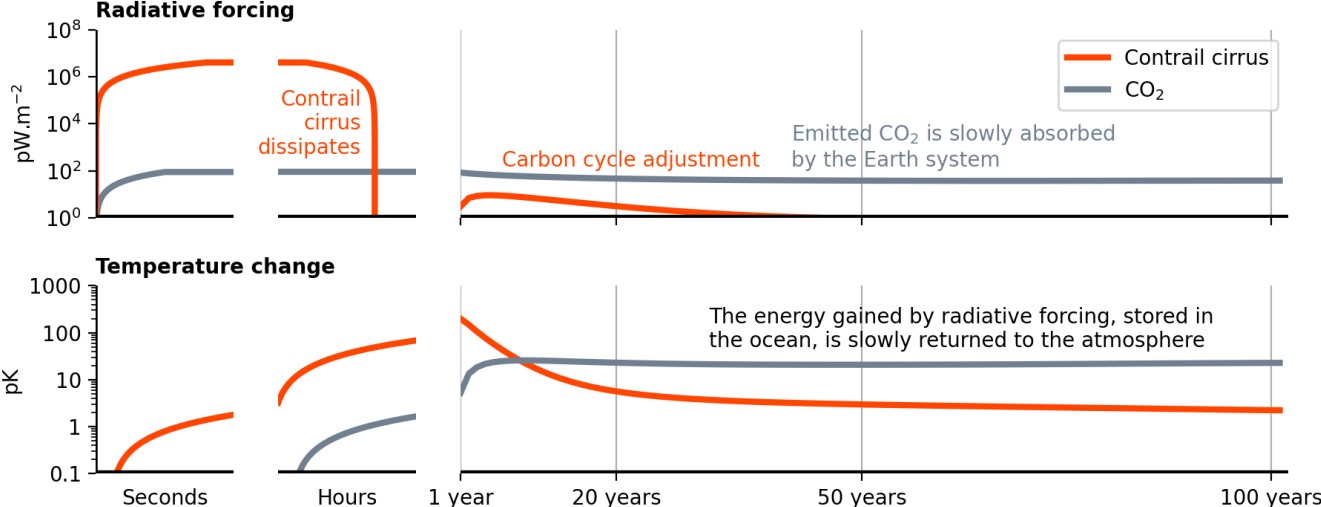

**Figure 1: Time evolution of (top row) globally averaged effective radiative forcing, in pW m$^{-2}$ ($10^{-12}$ W m$^{-2}$), and (bottom row) global surface temperature change, in pK ($10^{-12}$ K), for a flight that emits 53 tons of $CO_2$ (grey) in the atmosphere and forms a contrail (red) with an energy forcing of $2.9{\times}10^{10}$ J km$^{-1}$ over the whole 2450 km of its journey. Contrail efficacy is set to 0.37. Background $CO_2$ concentrations follow the SSP4-3.4 scenario.**

This discussion of time scales already implies that the behaviour of $CO_2$-equivalence metrics will depend on whether they are based on radiative forcing or temperature response, and whether they are integrated or endpoint metrics. Figure 2 compares the multiplicative factors associated with the AGWP, AGTP, and ATR using three time horizons, 20, 50, and 100 years, for the same flight presented in Figure 1. Multiplicative factors are here presented relative to the emitted $CO_2$, which is therefore given a value of 1. The values shown in Figure 2 are about half of those in Table 5 of Lee et al. (2021) because this study assumes a contrail efficacy of 0.37 compared to 0.42 in Lee et al. (2021), but mostly because Lee et al. (2021) computes the mean over all flights which gives an energy forcing of $3.6{\times}10^{10}$ J km$^{-1}$, while this study uses the median over

contrail-forming flights which is equal to $2.9\times10^{10}$ J km$^{-1}$. The distribution of contrail energy forcing per flown distance is strongly skewed by a few flights that form very strongly warming contrails, therefore the mean radiative forcing is shifted toward higher values than the median. Figure 2 includes the uncertainty in the climate response obtained using the Monte Carlo approach described above, which does not account for the uncertainties in contrail ERF and its efficacy, which are both large. According to the 5-95% confidence level in contrail cirrus ERF, as given by Lee et al. (2021) on the basis of the corresponding RF uncertainty, the contrail multiplicative factors could be between one-third and double the length shown in Figure 2.

The largest $CO_2$-equivalence metrics are obtained for a 20-year time horizon, regardless of the metric definition, and $CO_2$-equivalence tends toward 1 when time horizon increases. From a radiative forcing point of view, this is because the AGWP of contrails only slightly increases once the contrail has dissipated due to the carbon cycle feedback, while the AGWP of $CO_2$ keeps increasing with increasing time horizon. From a temperature response point of view, this is because the AGTP of contrails decays by two orders of magnitude after about 50 years. The AGWP20 and ATR20 of contrail cirrus are therefore large, ranging from 1.4 to 1.8 times the respective metric for $CO_2$. If these larger values were used in a practical setting, they would indicate a larger perceived benefit of contrail avoidance. Overall, AGWP and ATR yield greater multiplicative factors than AGTP, which places emphasis on the longer-term warming (Figure 1).

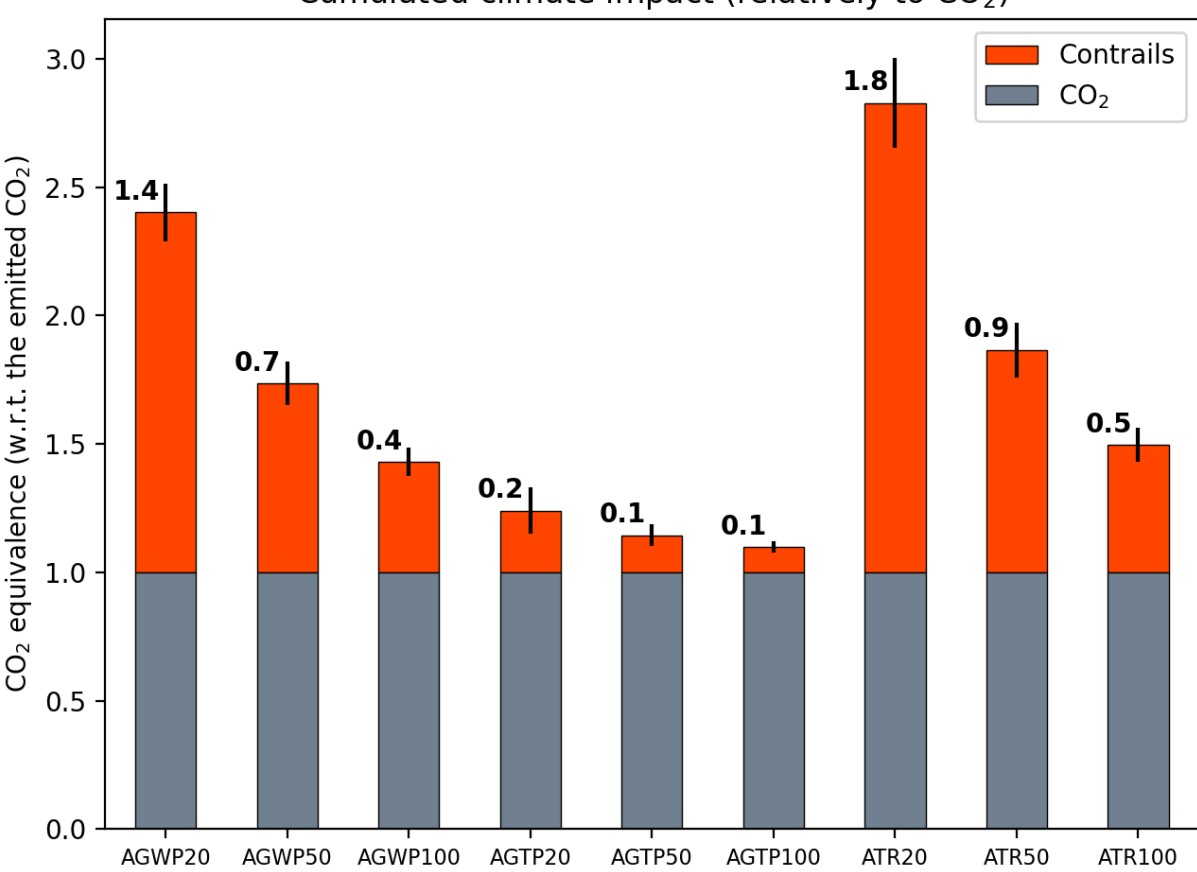

**Figure 2: CO₂-equivalence of a median North Atlantic contrail with a contrail efficacy of 0.37 (red) compared to the emitted CO₂ during the median flight (grey) when using Absolute Global Warming Potential (AGWP), Absolute Global Temperature Change Potential (AGTP), and Average Temperature Response (ATR) with time horizons of 20, 50, and 100 years. Error bars quantify the one-sigma uncertainty arising from the physical climate and carbon cycle of OSCAR, rather than uncertainties in the contrail radiative forcing and efficacy. The values at the top of each bar are the ratio of non-CO₂ to CO₂ effects for each metric choice.**

## 5. Contrail avoidance in the North Atlantic

### 5.1 Assessment of an idealised contrail avoidance scenario

The OSCAR model is now run for each of the 477,923 flights in 2019 considered by Teoh et al. (2022), including 260,854 contrail-forming flights. Figure 3 shows the absolute climate impact of those flights for the nine CO₂-equivalence metrics, for CO₂ and contrails and different flight categories. Overall, we estimate that the CO₂ emitted and the contrails formed by those flights will have warmed the climate by a total of 17.2 μK in 2039, 13.7 μK in 2069, and 14.1 μK in 2119, using AGTP. The 0.4 μK rise between 2069 and 2119 is due to CO₂ and the chosen background scenario, SSP4-3.4, which

assumes that $CO_2$ concentration decreases after 2080 (Meinshausen et al., 2019). Because the dependence between radiative forcing and $CO_2$ concentration is logarithmic (e.g., Etminan et al., 2016), emitted $CO_2$ has an increased radiative forcing when the atmospheric concentration decreases. In this case, this increase is greater than the decrease from the removal of $CO_2$ by the ocean and land surfaces, thus leading to a temperature increase from 11.0 µK in 2069 to 12.1 µK in 2119. During the same period, the warming originating from contrails decreases, from 2.7 µK to 2.1 µK. The ATR of $CO_2$ averages out the

small increases or decreases in temperature, being therefore almost constant at 11.7 <µK>, while the ATR of warming contrails decreases in time. Using AGWP, the impact of warming contrails is almost constant in time, because the contrail lifetime is much shorter than all considered time horizons, while the impact of $CO_2$ increases with time.

    Figure 3b shows that flights that form persistent warming contrails are responsible for 2.8 mW.m$^{-2}$.yr, 15.1 <µK> and 7.7 µK of warming from both $CO_2$ and contrails according to AGWP100, ATR100 and AGTP100. This corresponds to 71%,

72% and 55% of the warming from all flights, although flights that form persistent warming contrails only represent 44% of the dataset. For AGWP20, ATR20, and AGTP20, the contributions become 1.9 mW.m$^{-2}$.yr, 43.4 <µK> and 10.9 µK, respectively, corresponding to 87%, 90% and 63%, respectively, of the warming from all flights. The important decrease in the contribution from warming contrails with increasing time horizon indicates that the impact of flights forming warming contrails is largely dominated by the warming from these contrails, instead of the warming from $CO_2$. This is because the

warming from a persistent contrail decreases with time, relative to the warming from $CO_2$..

    In all panels, the impact of cooling contrails (Figure 3a, blue bars) or cooling flights and warming flights with cooling contrails (Figure 3b, blue and green bars) is limited, because the absolute total energy forcing from warming contrails is 23 times greater than that from cooling contrails. Total cooling from flights where contrail cooling exceeds $CO_2$ warming offsets at most 2% of total warming.

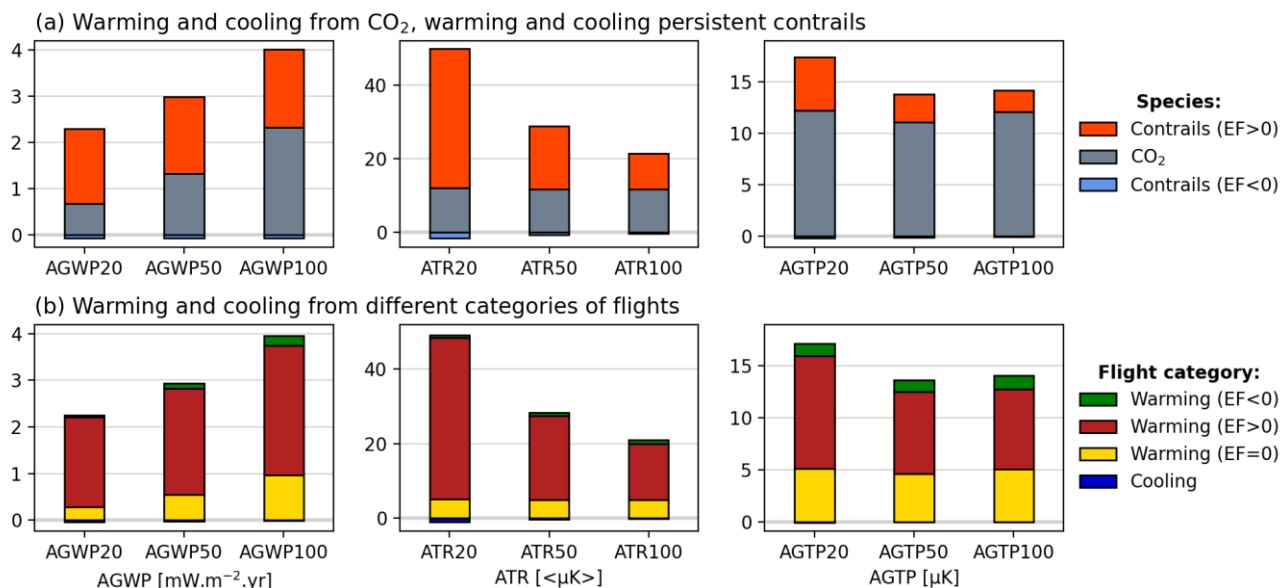

**Figure 3: Cumulative histogram of the warming or cooling by emitted $CO_2$ and persistent contrails formed by the 477,923 flights that crossed the North Atlantic sector in 2019 (Teoh et al., 2022) for nine $CO_2$-equivalence metrics. The first row (a) shows the decomposition into forcers: warming contrails (red), $CO_2$ (grey), and cooling contrails (blue). The second row (b) shows the decomposition into different categories of flights: warming flights that formed cooling contrails (green), warming flights that formed warming contrails (dark red), warming flights that did not form persistent contrails (yellow), and cooling flights (blue), including the effects of $CO_2$ for those flights. The corresponding numbers are presented in Tables S1 and S2 of the Supplementary Materials.**

OSCAR is used again to simulate the hypothetical rerouted flights. For each contrail-forming flight, we assume that it is possible to reroute the flight to avoid forming the persistent contrail. In practice, the increase in flight time and distance would depend on the size of the ISSR in which the contrail is formed and whether it is avoided horizontally or vertically. But what matters here is the additional $CO_2$ emitted by the rerouted flight. We set that additional $CO_2$ to 1% of the original flight emissions, which is in the range of increased fuel burn found in the case studies of Zengerling et al. (2023). In addition, fuel consumption and flight delays are both associated with large costs to airlines, so they might be unlikely to commit to larger increases. The additional $CO_2$ due to the rerouting would in practice depend also on the initial trajectory flown, ambient wind fields, and aircraft characteristics (Irvine et al., 2014). Rerouting is assumed to have been successful with persistent contrail formation being completely avoided, although it should be noted that this assumption is overly optimistic and cannot be currently guaranteed in any realistic setting.

Figure 4 shows the climate outcome of the rerouting, calculated here as the difference in AGTP100 between the rerouted and original flight, as a function of the contrail energy forcing of the original flight. AGTP100 is taken here as the reference because it is by definition equal to the temperature change after 100 years, so is easy to interpret physically. Rerouting, done within the assumptions above, leads to climate benefit for 69% of the contrail-forming flights (bottom right quadrant), for a total decrease in warming of 2.0 µK (−14% compared to total warming from all original flights). Rerouting the remaining 31% of the flights would damage climate, for a total increase in warming of 0.1 µK (+0.7% compared to total warming from all original flights). For those flights, the temperature change contribution by the rerouted flight is 5% larger than that of the original flight, either because the original flight was cooling (top left quadrant) or because the contrail formed had too weak positive radiative forcing to be worth the additional $CO_2$ under the 1% additional $CO_2$ assumption (top right quadrant). Illustrative case studies to clarify the results shown in Figure 4 are presented in Supplementary Materials.

It could be of interest to focus on flights in the bottom right corner of Figure 4 because their energy forcing might be sufficiently large to indicate with high likelihood that they do indeed warm the climate despite uncertainties in contrail EF. Such flights might be good candidates for "lower risk" contrail avoidance provided ISSR forecasts are reliable, such that there is a lower risk to reroute these flights than others. That subset of flights can be selected in various ways. For example, selecting flights with an energy forcing per contrail length larger than $10^{11}$ J km$^{-1}$ and whose climate benefit from contrail avoidance is 100 times larger than the climate damage from emitting 1% more $CO_2$ leaves 9.1% of contrail-forming flights. They contribute 1.48 µK to the initial AGTP100, which is reduced to 0.53 µK after rerouting. In other words, those 9.1% of flights represent 47% of the potential climate benefit of contrail avoidance. However, as will be shown in the next

subsection, the number of flights that match the selection criteria is a strong function of the choice of the $CO_2$-equivalence metric. There are many other options, which set the level of ambition and complexity of the rerouting, but also the climate benefit, which will be dependent on the $CO_2$-equivalence metric used.

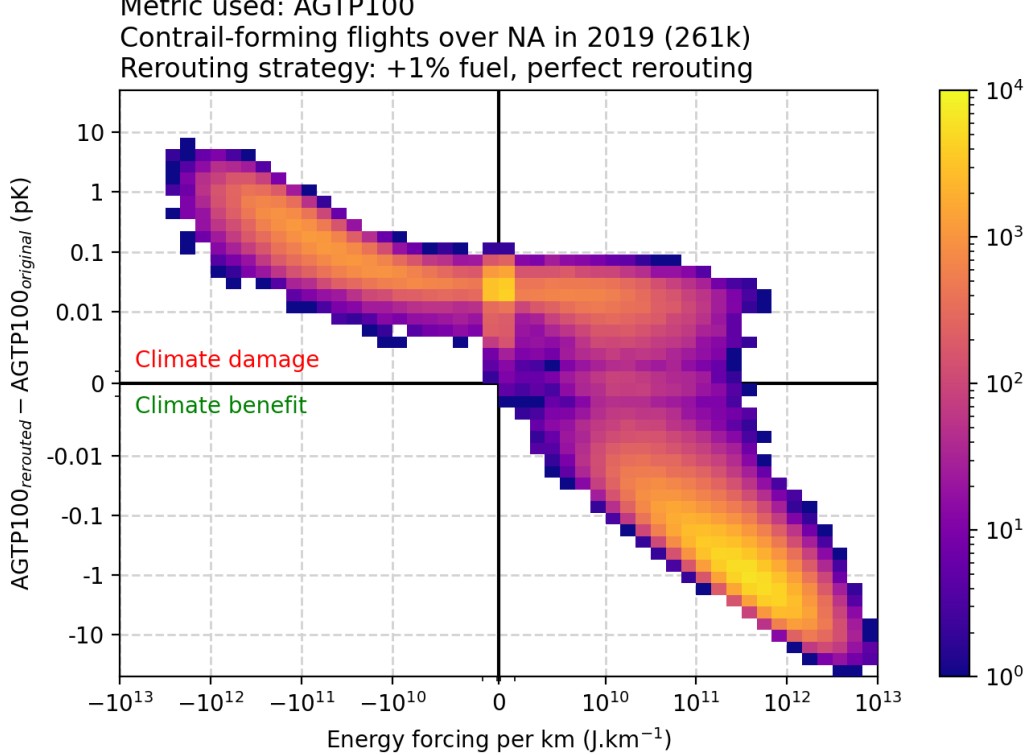

**Figure 4: Difference in AGTP100 between rerouted and original flights, in pK ($10^{-12}$ K), as a function of the energy forcing per persistent contrail distance of the original flight, in J km$^{-1}$. Flights are all the contrail-forming flights that flew over the North Atlantic in 2019. Colours indicate the number of flights for each combination, for a total of 260,854 contrail-forming flights. The rerouted flights emit 1% more CO₂ but do not form a contrail. Contrail efficacy is set to 0.37. The same figure expressed in other CO₂-equivalence metrics are in the Supplementary Materials.**

### 5.2 Sensitivity to the amount of additional CO₂ emissions

So far, the rerouting scenario assumed that contrails could be fully avoided by consuming 1% more fuel. Additional fuel consumption is now set to 0, 2 and 5%, to cover the range of estimated additional fuel burn from the case studies of Zengerling et al. (2023). Results are shown in Figure 5 for the nine $CO_2$-equivalence metrics. Emitting more $CO_2$ forces more flights into the climate damage category for all metrics, as defined in the previous sub-section. Focusing on AGTP100, the effect is sizeable in terms of number of flights: at 5% additional $CO_2$, rerouting damages climate for 43% of contrail-forming flights, 12 percentage points more than the assumption with 1% additional $CO_2$ (31%). In the scenario where no

additional $CO_2$ is emitted, the number of reroutings that damage climate is decreased by 11 percentage points compared to the 1% additional $CO_2$ scenario, falling to 20% of contrail-forming flights. These flights damage the climate because a cooling contrail is avoided. The number of "lower risk" reroutings is 5 times larger when no additional fuel is emitted compared to the +1% fuel scenario. This is because our definition of "lower risk" reroutings relies on a maximum amount of additional fuel, and this condition is always met when no additional fuel is emitted.

Choosing a shorter time horizon enhances the perceived climate benefit but underestimates the long-term climate damage, as shown in Figure 5b. Using AGTP20 and the 1% additional $CO_2$ scenario, 11% fewer flights fall in the climate damage category than with AGTP100. Additionally, the climate benefit from all reroutings depends slightly on the fuel scenario, especially for $CO_2$-equivalence metrics that put more weight on the impact of contrails, such as ATR20. However, "lower risk" reroutings depend much more on the scenario. For example for AGTP100, the number of "lower risk" reroutings is

almost 0 in the +5% scenario, indicating that although the perceived climate benefit is rather high, some reroutings could in fact lead to climate damage because of the uncertainties in contrail EF.

    For a given $CO_2$ scenario and for most contrail-forming flights, all $CO_2$-equivalence metrics agree that rerouting would benefit climate. Disagreements between $CO_2$-equivalence metrics happen for about 10% of flights, which form low energy contrails that do not contribute much to climate damage anyway. The range of contrail EF where at least one $CO_2$-

equivalence metric gives a different climate outcome of rerouting than the other metrics depends on the flight and $CO_2$ scenario. For the median flight with 1% additional $CO_2$, that range spans 7% of the contrail-forming flights, from the 24th to the 31st percentiles. This range represents very low-energy contrails, with EF per flown kilometre from $1.7 \times 10^8$ to $3.1 \times 10^9$ J km$^{-1}$. However, the perceived climate benefit as shown in Figure 5b depends greatly on the $CO_2$-equivalence metric.

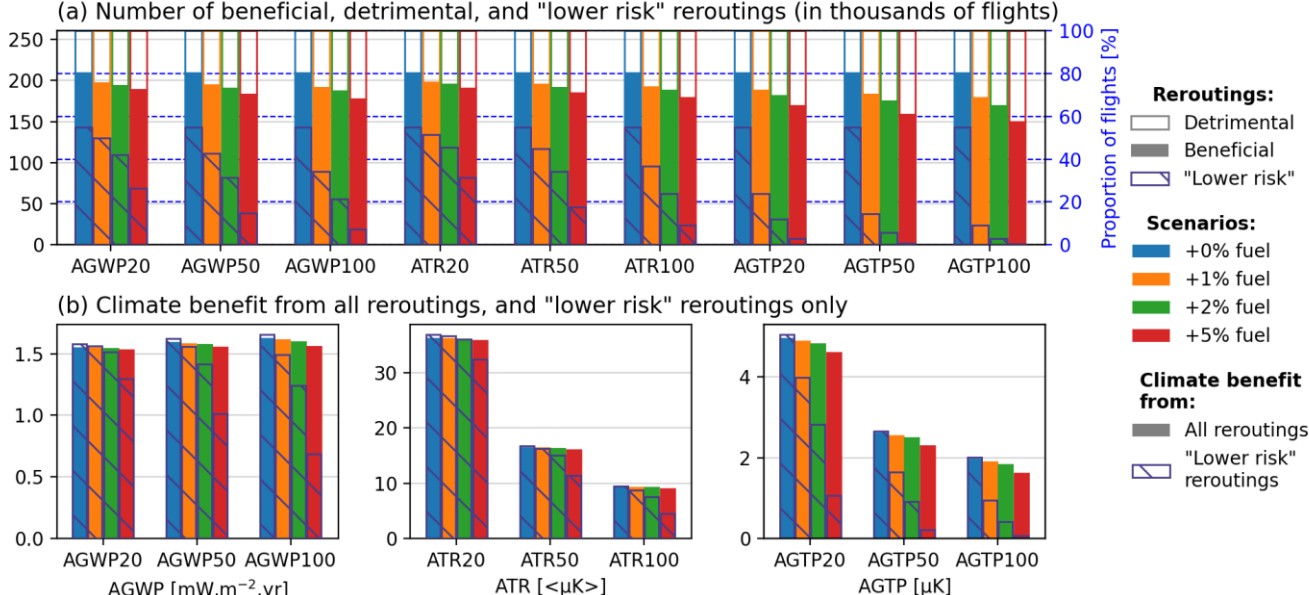

**Figure 5: (First row) Number of flights flown over the North Atlantic in 2019, for a total of 260,854 contrail-forming flights, for which rerouting leads to climate benefit (colours) or climate damage (white) according to nine $CO_2$-equivalence metrics and depending on the additional $CO_2$ emitted to reroute. Colours indicate the amount of additional $CO_2$ emitted to reroute: +0% (blue), +1% (orange), +2% (green) or +5% (red). Rerouting is assumed to be successful, so the rerouted flight does not form a contrail, and a contrail would have formed on the original route. "Lower risk" reroutings (see text) are marked with hatches. (Second row) As first row but showing the climate benefit from rerouting all contrail-forming flights. The corresponding numbers are presented in Table S3 of the Supplementary Materials.**

### 5.3 Sensitivity to partial contrail avoidance

The results presented in Figure 5 are idealised because the additional $CO_2$ is independent of the distance over which a contrail is formed. It is plausible that 1% additional $CO_2$, and indeed even 5%, would not allow the complete avoidance of some contrails. Determining the best avoidance strategy for a given flight requires accurate forecasts of meteorological conditions for that flight, which is out of the scope of this study. There is no simple relationship between additional $CO_2$ and contrail EF: ISSRs could be orthogonal to the plane trajectory so difficult to avoid, or conversely vertical avoidance might be very easy, irrespective of contrail EF. There might also be cases where two different routes have very different ISSR occurrence but identical fuel burn.

But to illustrate the impact of partial contrail avoidance, we assume here that the longer the contrail, the more difficult it is to completely avoid. A rerouting efficiency factor is therefore calculated as the ratio of the contrail length to total distance flown for the original flight. For each rerouting, contrail energy forcing is inversely scaled by this rerouting efficiency factor, which is less than the 100% efficiency assumed in the full contrail avoidance scenario. So, rerouting efficiency is set to 0% for a contrail that is as long as the flight length, meaning that the rerouting fails to reduce the energy forcing of the contrail.

Rerouting efficiency will tend toward 100% for contrails that are very small compared to the length of the flight, meaning that the contrail is fully avoided. For the contrail-forming flights flown over the North Atlantic in 2019, we calculate that the

average rerouting efficiency for all reroutings would be 71%, and the 5th and 95th percentiles are 31% and 95%, respectively. Table 1 shows the number of reroutings beneficial to climate and the absolute benefit of all reroutings (both beneficial and detrimental to climate) as a function of the $CO_2$-equivalence metric. For the sake of simplicity, the +1% fuel scenario is used for all reroutings, and results for full and partial avoidance are compared. The total decrease in warming in 100 years (i.e., using AGTP100) is 1.02 μK, a reduction of 46% compared with assuming a rerouting efficiency of 100% for all flights,

where total warming was decreased by 1.90 μK. That reduction is larger than the reduction in average rerouting efficiency, which is only 29% (from 100% to 71%), because high-energy contrails are often longer than low-energy ones. Therefore, their rerouting efficiency is lower, although they contribute more to the absolute benefit of the rerouting. In contrast, the number of reroutings leading to climate benefit in terms of AGTP100 decreases by only 3%, from 179,226 to 174,413. That is because reducing the climate impact of strongly warming contrails leads to a climate benefit even when that reduction is

incomplete.

The relative decrease in climate benefit depends only slightly on the $CO_2$-equivalence metric. The number of reroutings that benefit climate does not change much in this partial contrail avoidance calculation, ranging from −1% to −3%. This indicates that the previous finding that qualitative decision making is almost independent on the choice of $CO_2$-equivalence metric remains valid. Moreover, the absolute benefit is reduced by around 45% for all $CO_2$-equivalence metrics. As we showed

earlier in this Section, this is because a large part of the absolute benefit comes from flights for which the impact of the additional emitted $CO_2$ is much lower than the impact of the avoided contrail. For these flights, $CO_2$ is of secondary importance, therefore the $CO_2$-equivalence metrics mostly quantify the climate outcome of contrail avoidance, which is reduced in a partial avoidance scenario.

**Table 1: Number of flights flown over the North Atlantic in 2019, out of a total of 260,854 contrail-forming flights, whose rerouting is beneficial to climate, and absolute benefit of all reroutings (beneficial and detrimental to climate), assuming that contrail formation is fully or partially avoided (see text), as a function of $CO_2$-equivalence metric.**

| $CO_2$-equivalence metric | Number of flight reroutings beneficial to climate | | Absolute climate benefit of all reroutings | |
| --- | --- | --- | --- | --- |
| | Full avoidance | Partial avoidance | Full avoidance | Partial avoidance |
| AGWP100 | 191,444 | 188,754 (−1%) | 1.61 mW m$^{-2}$ yr | 0.88 mW m$^{-2}$ yr (−45%) |
| AGWP50 | 194,248 | 191,850 (−1%) | 1.58 mW m$^{-2}$ yr | 0.87 mW m$^{-2}$ yr (−45%) |
| AGWP20 | 196,910 | 194,824 (−1%) | 1.54 mW m$^{-2}$ yr | 0.85 mW m$^{-2}$ yr (−45%) |
| AGTP100 | 179,226 | 174,413 (−3%) | 1.90 μK | 1.02 μK (−46%) |
| AGTP50 | 183,326 | 179,203 (−2%) | 2.55 μK | 1.38 μK (−46%) |

| | | | | |
|---|---|---|---|---|
| AGTP20 | 187,828 | 184,618 (−2%) | 4.88 μK | 2.66 μK (−45%) |
| ATR100 | 192,252 | 189,661 (−1%) | 9.27 <μK> | 5.08 <μK> (−45%) |
| ATR50 | 195,051 | 192,684 (−1%) | 16.3 <μK> | 8.98 <μK> (−45%) |
| ATR20 | 197,878 | 195,834 (−1%) | 36.1 <μK> | 19.9 <μK> (−45%) |

## 5.4 Sensitivity to contrail efficacy

Contrail efficacy has been set so far to 0.37. To study the sensitivity of our results to that choice, contrail efficacy is varied from 1 to 0.05 by considering six values:

- 0.21, 0.31 and 0.59, which are the estimates from Bickel (2023), Rap et al. (2010) and Ponater et al. (2005), respectively, for the total contrail efficacy.

- 1.0, which is not supported by climate modelling but corresponds to a focus on contrail radiative forcing, rather than effective radiative forcing.

- 0.05 and 0.1, which are very low efficacies that might be possible if the lower end of the Bickel et al. (2020) confidence interval for the ERF-to-RF ratio was multiplied by an ERF-based efficacy of 0.38 (Bickel, 2023).

Figure 6a shows that the number of reroutings beneficial to climate does not change drastically between different choices of contrail efficacy. For contrail efficacies between 0.05 to 1, the maximum difference is 25% for AGTP100 for a contrail efficacy of 0.05. This implies that contrail efficacy plays a minor role in the decision making of a rerouting: in the worst case, 75% of the reroutings that benefit climate remain beneficial. This again comes from the fact that a warming contrail is, most of the time and for all the $CO_2$-equivalence metrics we investigated, a few orders of magnitudes more warming than the additional $CO_2$ emitted to avoid it.

In contrast, the perceived climate benefit associated to a rerouting depends highly on contrail efficacy. Figure 6b shows that the impact of contrail efficacy on the quantification of the climate benefit of reroutings is much larger. Climate benefit is more than 2.5 times larger for a contrail efficacy of 1 than for an efficacy of 0.37, and about 10 times smaller for a contrail efficacy of 0.05. It should be noted that this impact does not depend much on the $CO_2$-equivalence metric, because the climate benefit of very high-energy contrails is much greater than the climate damage by the additional emitted $CO_2$. Therefore, changing the contrail efficacy by a given factor changes the absolute climate benefit by about the same factor, for all the considered metrics. However, this is only true when considering rerouted flights together, not on a flight-to-flight basis.

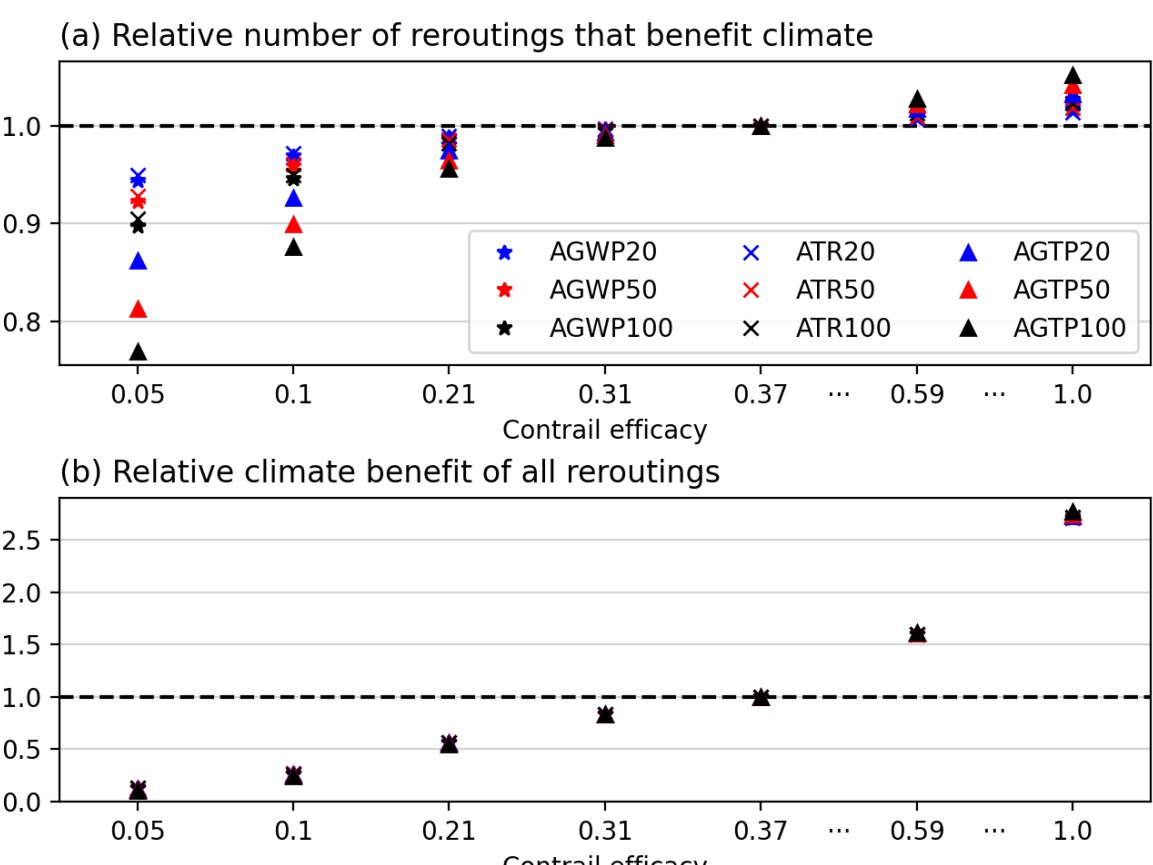

**Figure 6: Relative number of reroutings that benefit climate (a) and relative climate benefit of all reroutings (b), for the nine $CO_2$-equivalence metrics considered in this study (colours and symbols) and as a function of contrail efficacy. Additional $CO_2$ emitted is 1%, and contrail avoidance is assumed to be complete. Values are normalised to the case where contrail efficacy is 0.37.**

## 6. Conclusion

The different time scales of $CO_2$ and contrail cirrus radiative forcing and temperature response cause different behaviour of $CO_2$-equivalent metrics. Time-integrated metrics defined on a short time horizon, like AGWP20 or ATR20, put more weight on contrail cirrus, while endpoint metrics on a long time horizon, like AGTP100, put more weight on $CO_2$. This behaviour, which had already been noted (e.g., Deuber et al., 2013; Irvine et al., 2014; Grewe et al., 2014), means that the choice of metric definition and time horizon can have implications for decision making related to contrail avoidance. Using an analysis of potential contrail avoidance strategies in the North Atlantic flight corridor, this study shows that the decision whether to reroute a flight or not is generally independent of the choice of $CO_2$-equivalent metric. This is because when a persistent warming contrail is formed, it is often orders of magnitude more warming than the potential additional emission of $CO_2$ to

avoid its formation. In many of these situations, all the investigated metrics indicate that the flight should be rerouted, although the weight put on contrails is not the same.

The lack of consensus on what is a suitable or the correct $CO_2$-equivalence metric is therefore not an obstacle to implementing contrail avoidance policies. However, the quantification of how much additional $CO_2$ can be emitted on a rerouted flight and the resulting climate outcome of a rerouting both depend on that choice. For a flight for which all metrics agree that a rerouting would benefit climate, time-integrated metrics with short time horizons, like AGWP20 or ATR20, calculate a much greater apparent climate benefit than endpoint metrics with long time horizons like AGTP100, which in

many situations indicates that the benefit is moderate. Assuming that additional $CO_2$ emissions are fixed percentages of original emissions, or an increasing function of contrail length does not change these findings. The same conclusion is also drawn when considering sensitivity to contrail efficacy: the number of reroutings that benefit climate is a weak function of contrail efficacy, but the size of the climate benefit is very dependent on contrail efficacy.

    This study used the contrail energy forcing distribution from Teoh et al. (2022) computed with CoCiP (Schumann et al.,

2012) as radiative forcing input to the OSCAR compact Earth system model (Gasser et al., 2020). Rerouting was represented in a simplified way because the knowledge of flight trajectories and meteorology has been lost into the contrail energy forcing distribution. Future studies should investigate more realistic cases, considering the meteorology and particularly the horizontal and vertical sizes and humidity profiles of ISSRs. However, our work shows that even a partially avoided contrail can be beneficial for the climate if the contrail is sufficiently warming (e.g., with a contrail energy forcing per contrail length

greater than $10^{11}$ J km$^{-1}$), and that many flights have formed such contrails over the North Atlantic in 2019 (30% of all flights for the latter condition). The contrail climate impact is often orders of magnitude greater than the impact of the additional emitted $CO_2$, so a rerouting can still be effective if it leads to the formation of much weaker contrail. Moreover, this study has assumed that contrail avoidance translates into unchanged or increased fuel burn and $CO_2$ emissions. In fact, current flight trajectories are predominantly optimised to minimise total cost (including, but not limited to, fuel burn, airspace

charges, and staff costs), so there is a possibility that rerouting flights to minimise in addition the climate impact by avoiding contrails could lead to a reduction in overall fuel burn for some flights (Dalmau et al., 2018).

    This study focuses on the North Atlantic, so its results do not necessarily carry over to other regions which may have different characteristics, for example in the diurnal variation of air traffic, the probability of encountering ISSRs and the average altitude of flights, as well as the practicality and operational constraints of rerouting due to the volume of air traffic.

However, the large difference between contrail warming and that of the additional emitted $CO_2$ seen for the most warming flights is likely a defining feature of such flights in any region, so the fundamental results are expected to remain valid.

    Implementing a contrail avoidance scheme is likely to be complex because of the difficulties in predicting ISSRs (Gierens et al., 2020), or the large uncertainties in calculating the radiative forcing of contrails (Wilhelm et al., 2021), or operational challenges with air traffic management. Sausen et al. (2023) found in a case study of contrail avoidance over the Netherlands

that persistent contrails were not observed in about half the cases when they were forecast, which indicates the risk of unnecessary diversions. Lee et al. (2021) found a 5-95% confidence level for contrail cirrus radiative forcing of 33 to 189

mW m$^{-2}$. By contrast, the $CO_2$ forcing is uncertain by only $\pm 10\%$ (31 – 38 mW m$^{-2}$). This study considered uncertainties originating from the OSCAR climate model, which are comprised within a 10% range, but did not account for uncertainties in contrail radiative forcing or ISSR prediction in the ERA5 reanalysis. On a flight-by-flight basis, rerouting would need to use operational weather forecasts rather than a reanalysis. However, such forecasts are known to have a greater uncertainty. So, the jury is still out as to whether contrail avoidance has in practice the potential to substantially reduce the total radiative forcing of aviation, and whether that reduction can be validated. One way to avoid dealing directly with uncertainty could be to target the subset of strongly warming contrails whose energy forcing is sufficiently strong to remain positive despite large contrail uncertainties, assuming these can be adequately forecast. But our results suggest that the number of flights selected in that way strongly depends on the choice of $CO_2$-equivalence metrics. A cautious approach to select such flights could be to use AGTP100 to maximise the probability that the rerouting does indeed benefit climate. The next step is therefore to link the present analysis to estimates of contrail radiative forcing uncertainty and to use real cases of contrail avoidance reroutings to quantify the potential contrail energy forcing reduction with the corresponding additional emitted $CO_2$.

Which $CO_2$-equivalence metric works best for contrail avoidance? This study offers some insights. In terms of choice of time horizon, rerouted flights can be perceived to be a climate benefit using short time horizons. However, an increasing proportion of these become climate damaging as the time horizon is increased, as shown in Figure 7. Indeed, it has been noted that the use of GWP20 is equivalent to use a discount rate of more than 10% (Sarofim and Giordano, 2018; Mallapragada and Mignone, 2020), which is much higher than the discount rate of 4 to 5% typically assumed for designing cost-effective mitigation scenarios using Integrated Assessment Models (Emmerling et al. 2019). Furthermore, it can be argued that aviation would have to be at an advanced stage of decarbonisation, at least in some regions, to justify using a shorter time horizon than the 100 years used for surface emissions from other economic sectors. In terms of choice of metric definition, AGWP and ATR, which both involve a time integration over the time horizon, behave very similarly, both qualitatively and quantitatively. AGWP has the advantage over ATR of being less subject to uncertainties, whereas AGWP is less comprehensive than ATR in terms of the consideration of climate impacts. AGTP behaves differently from the other two metrics, and generally predicts less positive climate outcomes of contrail avoidance. But it has the advantage of measuring surface temperature change, which is directly relevant to the warming target of the Paris Agreement.

Until such time that a political decision has been made, any proposed contrail avoidance schemes should, in addition to considering the many scientific uncertainties inherent in such schemes, consider a range of metrics and time horizons, such as those used in this study, to assess the robustness of rerouting decisions and to quantify the actual climate benefit. Such a recommendation has already been made (Levasseur et al., 2016; Cherubini et al., 2016; Jolliet et al., 2018) and applied (e.g., Tanaka et al., 2019; Reisinger et al., 2017; Tibrewal et al., 2020) by the Life Cycle Impact Assessment community, which uses by consensus GWP100, GWP20, GTP100, and GTP20 in their assessments.

**Data availability**

CO₂ emission and contrail energy forcing data were made available from Roger Teoh (roger.teoh15@imperial.ac.uk; Teoh et al., 2020). The source code of OSCAR is available at https://github.com/tgasser/OSCAR. Additional scripts and data are available upon request from the corresponding author.

**Author contribution**

AB and NB conceptualised and conducted the study, and prepared the manuscript. All the authors discussed and commented on the manuscript.

**Competing interests**

The authors declare that they have no conflict of interest.

**Acknowledgements**

AB, OB, and NB acknowledge the support of the French Ministère de la Transition écologique et solidaire (N° DGAC 382 N2021-39), with support from France's Plan National de Relance et de Résilience (PNRR) and the European Union's NextGenerationEU. K.T. benefited from State assistance managed by the National Research Agency in France under the

Programme d'Investissements d'Avenir under the reference ANR-19-MPGA-0008.

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
