# Peer review of "The importance of an informed choice of CO2-equivalence metrics for contrail avoidance"

_EGUsphere, 2024_

## Referee Comment (RC2)

Review of the manuscript "The importance of an informed choice of $CO_2$-equivalence metrics for contrail avoidance" by A. Borella et al. (egusphere-2024-347)

**Recommendation**

This is an appropriate and timely contribution to the rapidly emerging discussion on how to assess the climate effect of contrail avoidance measures. It certainly contains a lot of insightful information for scientists and stakeholders alike. The strengths of the paper are to be found in the clearly presented guiding ideas that highlight and illustrate the main problems when quantifying the climate impact gain of contrail avoidance. I am especially pleased with the inclusion of surface temperature change efficacy in the metrics calculations, as this aspect has been disregarded all too often in previous respective assessments. Overall, the conceptual framework and the metrics calculations seem fully adequate to me.

However, I also perceive a number of shortcomings in the presentation of the results, both with respect to a precise description of the methods and to a coherent illustration of technical details. A particular weakness is insufficient explanation of some figures and a lack of hints to the additional material in the supplement. Transferring text from the figure captions and/or from the supplemental material to the main text may be *one* idea for amendment. As the paper deals with a topic of high socio-economic relevance, it is essential to avoid creating any kind of misunderstandings and confusion. Therefore, I encourage the authors to ensure maximum precision all through the text, also when describing the details. I will give a number of major and minor points where I see room for improvement in this respect.

I recommend the paper to be accepted for publication in ACP after those points have been addressed.

**Major Concerns**

1. Formulations in the abstract are generally sound and transparent, except for the final statement that "the choice [of metrics] is ultimately political". I think that could be expressed more clearly, e.g. "the choice of metrics implies to lay a focus on climate impact reduction on shorter or longer time scales, which is ultimately a political decision". In case that is what you meant to say. Any ascertainment should then be naturally emerging from the discussion in the conclusions section.

2. The discussion of the results presented in Figure 3 is much to brief and poor guidance is given to the reader to relate the text interpretation to certain figure features. This is emphasized by the fact that the text gives contributions in %, while the figure provides the AGWP/ATR/AGTP values in absolute units. E.g., one feature needing explanation is the $CO_2$ ATR independence from the time horizon, while there is a respective dependency for the end-point metric AGTP. Is there an obvious reason why the relatively large number of contrails with negative EF (line 155) contribute so little to ATR/AGTP? Please realize that these results need to be plausible also to non-experts, and adjust/extend the text accordingly.

3. In Figure 5, again, the discrepancy between the relative numbers (%) discussed in the text and the absolute numbers (thousands of flights) in the figure makes it difficult to follow the reasoning. I urgently recommend to insert a complementing scale of percentage-numbers along the right y-axis. The whole discussion, however, seems to be rather fuzzy here. The sentence "The number of "lower risk"

reroutings is 550% larger" (hardly comprehensible in itself) seems to be the only interpretation of the second row of Figure 5, which is clearly insufficient to convince the reader of anything. Please, extend the discussion and refer to exactly the columns you are interpreting.

4. Some confusion already exists in the (sparse) literature on the choice of numbers for certain efficacy parameters in the "radiative forcing" (RF) framework and in the "effective radiative forcing" (ERF) framework, and I feel that your paper could help to amend respective inconsistencies. The paper correctly distinguishes between a contrail "total efficacy" (line 180), which results as the product of an ERF/RF factor (originating from rapid radiative adjustments induced by the contrail forcing) and the ERF related efficacy (line 178) that originates from surface temperature induced radiative feedbacks. You have chosen a value of 0.35 for the total efficacy by multiplying the ERF/RF factor (0.35) from Bickel et al. (2020) with a value of 1 for the ERF related efficacy "for lack of a better estimate". Yet, there is an estimate of the ERF related efficacy and it's given in line 377! While there may be a good reason to choose the values as you do, any source of misunderstanding in the context should be eliminated.
When it comes to uncertainty considerations in section 5.3, the discussion starts with the sentence "Contrail efficacy has been set so far to the best estimate of 0.35 from Bickel et al. (2020)." This wording is confusing, because the Bickel et al.'s value is actually an ERF/RF factor – as you correctly introduce in section 3. You probably meant to repeat the section 3 definition in a nutshell, but actually this combination of ERF/RF factors and efficacies emphasizes a respective simplifying approach by Lee et al. (2021), who calculated a best estimate for the contrail cirrus ERF/RF factor (value 0.42 – as referenced in line 222) from three individual model results of which two (Ponater et al., 2005, and Rap et al., 2010) had rather provided a total efficacy value. It should be easy to avoid such simplifications in the context of your paper.
An analogous criticism applies to your choice of "contrail efficacy" values 0.23 and 0.51 (line 377) referring to the confidence interval given by Bickel et al. (2020) – but again that was for their ERF/RF factor. For choice of a lower bound for contrail total efficacy, the ERF related efficacy best estimate from Bickel (2023), i.e., 0.38, is combined with the lower estimate of the ERF/RF factor from Bickel et al. (2020). This is largely okay in a qualitative sense, but it neglects that Bickel (2023) also provides a total efficacy value in his Table 4.3., i.e. 0.21, by combining their 0.38 value for ERF related efficacy with the ERF/RF factor within their own model framework – which is more consistent. Still, given that Bickel's (2023) total efficacy value (0.21) will also have an uncertainty (which is not quantified in his thesis), your estimated parameter range for total efficacy appears quite reasonable to me.

5. The total efficacy as chosen in the paper (0.35) "is assumed to apply for all contrails" (line 182). This may well be acceptable for the conceptual targets of this paper, yet it implies that the rapid radiative adjustments (controlling ERF/RF), as well as the surface temperature induced radiative feedbacks, are independent of where and when a contrail develops. This is, obviously, a very unlikely assumption. The authors appear to be well aware of this fact, as (line 389) they write that the climate benefit from contrail avoidance, when calculated accordingly, is only valid "when considering rerouted flight together, not on a flight-to-flight basis". This overall assessment is likely correct (or, at least, tenable), but I feel that an extra warning is necessary here, in order to avoid the impression that RF (or the

energy forcing, EF) of an individual contrail (or flight) can safely be converted to ERF by using a uniform factor derived from global considerations.

6. There is a further potential source of error when introducing efficacy parameters in metrics calculations, as the CoCiP model primarily uses energy forcing (EF) to assess the climate impact of individual flights (Schumann et al., 2012; Teoh et al., 2020), which can then be used to calculate a global instantaneous radiative forcing ($RF_{inst}$). However, global ERF/RF, climate sensitivity and efficacy parameters (e.g. Bickel et al., 2020; Lee et al., 2021; Bickel, 2023) are usually given on the basis of the stratosphere adjusted radiative forcing ($RF_{adj}$). Switching between $RF_{adj}$ and $RF_{inst}$ makes not much of a difference for contrails (Dietmüller et al., 2016, their Table 2), but strongly matters when calculating the tradeoff with the radiative effect induced by additional $CO_2$ emissions. I presume that, concerning your paper, OSCAR uses additional fuel consumption (rather than a $CO_2$ EF provided by CoCiP) to calculate the $CO_2$ $RF_{adj}$. I deem it important that the text should not leave open any doubt on this (line 160), as otherwise a combination with the published global ERF/RF factors (and efficacies) would be inconsistent (see efficacy fluctuations for various RF definitions, as reported by Richardson et al., 2019, their Table 1).

**Specific remarks/recommendations**

Pg. 1, l. 17: "… defined here as nine combinations of different definitions …" sounds somewhat strange. Perhaps "… represented here as nine combinations …"?

Pg. 1, l. 19: I recommend to write "Under an idealized scenario where …" to avoid any misunderstanding.

Pg. 2, l. 33: As there is no extra definition section of radiative forcing in this paper, I recommend to insert another sentence here (before "According to"): "The climate effect of $CO_2$ and non-$CO_2$ contributions has usually been compared in terms of several radiative forcing parameters (e.g., Ramaswamy et al., 2018) or dedicated metrics (e.g., Fuglestvedt et al., 2010).

Pg. 4, l. 103: "It is impractical …", this statement demands a rationale or a reference.

Pg. 6, l. 170: "… 1726 simulations …"; Why is this selection necessary? Is this relevant for the outcome?

Pg. 6, l. 175: A personal remark at this point: You choose to label the ERF related efficacy (line 178) that originates from surface temperature induced radiative feedbacks as the "Ponater efficacy". While I feel flattered by this labelling, the term might still be misleading, as the cited reference by Ponater et al. (2021) deals with both the RF and the ERF related efficacy parameter and their interrelation, but without giving numbers. Different definitions of efficacy were introduced by Hansen et al. (2005), but the first dedicated calculation of the ERF related efficacy for contrails was given by Bickel (2023), yielding a value of 0.38, as is mentioned in your paper but only later (line 377).

Pg. 6, l. 180: Please, add references to Hansen et al. (2005) and Richardson et al. (2019).

Pg. 6, l. 182: "… the same factor is assumed for all contrails." Does this mean that each contrail EF is multiplied with the assumed contrail efficacy? Is this relevant for OSCAR or does that model only receive global mean contrail RF as an input? – See major concerns.

Pg. 6, l. 187: Recommended modification: "… are illustrated in an idealized but representative way by Figure 1, …"

Pg. 7, l. 202-204: This sounds as if the ocean surface layer absorbs more or less instantaneously the whole energy provided by any forcing and then returns the temperature signal slowly back to the atmosphere, but such a notion would be inconsistent with the known large ocean heat capacity. However, I presume that's not what you mean. Please make it plain that the time limit for the heat transfer from the atmosphere to the ocean is prescribed in your scenario through the length of the "pulse" for which the contrail forcing lasts.

Pg. 8, l. 229: Lee et al. (2021), Appendix E, explicitly states: "When calculating the contrail cirrus ERF, the error range given refers to the error range of contrail cirrus RF and not ERF." Hence, I recommend to modify your formulation to "… confidence level in contrail cirrus ERF, as given by Lee et al. (2021) on the basis of corresponding RF uncertainty considerations."

Pg. 9, l. 244: Choice of the headlines in this section 5 seems a bit incoherent to me. One idea could be to title the subsection here as "5.1 Assessment of an idealized contrail avoidance scenario"; then replace the heading of what is now sub-section "5.1 Sensitivity to additional $CO_2$" by "5.2 Sensitivity to the amount of additional $CO_2$ emissions"; and then replace the heading of what is now "5.2 Imperfect contrail avoidance" by "5.3 Sensitivity to imperfect contrail avoidance". The 5.3 heading could remain as it is (but then as 5.4). Please, consider this suggestion as a potential improvement.

Pg. 11, l. 290: "Figure 5 shows …", seems bad wording to me as Figure 5 is in fact dealt with only in the following sub-section. "As will be shown in the next sub-section" may be an improvement. But see major concerns (above) with respect to Figure 5.

Pg. 12, l. 305: "…into the climate damage category, as defined in the previous sub-section"

Pg. 12, l. 310: "550% larger"; I fail to relate this statement to any feature in Figure 5, nor do I understand what the statement actually means (larger than what?). But see major concerns above.

Pg. 14, l. 335: "There is no simple relationship between addition $CO_2$ and contrail EF": True indeed! The statement could be moved to the begin of the sub-section to motivate why "The results presented in Figure 5 are idealised" (line 332), despite the fact that the additional fuel increment is varied in that figure.

Pg. 14, l. 342: "… the rerouting fails to reduce the energy forcing …" Completely? Why? I fail to perceive the logic in this assumption.

Pg. 14, l 344: "… the average rerouting efficiency is 0.71 …"; How is this number calculated? And how is it, then, used to modify the original results presented in Figure 4? Is this addressed in the following paragraph?

p. 14, l. 351: "In contrast, … temperature rise in 100 years … by only 3%". Which line in Table 1 are you referring to here, which numbers are you comparing? Is "warming in 100 years" (l. 342) referring to something different than "temperature rise in 100 years" (l. 352)? Please, be precise and refer to the actual metric you are addressing.

p. 14, l. 356: "… the absolute benefit comes from flights for which the impact of the additional emitted $CO_2$ is much greater than the impact of the avoided contrail"; the statement strikes me as illogical. What do you mean by this?

p. 15, l. 371: "Contrail efficacy has been set to the best estimate of 0.35 from Bickel et al. (2020). This statement is confusing as Bickel et al. provide an ERF/RF factor. Please, consider the efficacy related part of the major concerns above.

p. 16, l. 396: In some contrast to the results section, formulations in the conclusions section are generally sound and precise.

p. 16, l. 401: Some more references to previous work would be fine here. Immediately to my mind come Deuber et al. (2013) and Irvine et al. (2014).

**References:**

Bickel, M., et al., 2020: Estimating the effective radiative forcing of contrail cirrus, J. Clim. 33, 1991-2005.

Bickel, M., 2023: Climate impact of contrail cirrus, DLR-Forschungsbericht 2023-14, PhD Thesis, Ludwig-Maximilians-Universität München, https://edoc.ub.uni-muenchen.de/view/autoren/Bickel=3AMarius=3A=3A.html

Deuber, O., Matthes, S., Sausen, R., et al., 2013: A physical metric-based framework for evaluating the climate trade-off between CO2 and contrails – the case of lowering aircraft flight trajectories, Environ. Sci. Policy 25, 176-185.

Dietmüller, S. et al., 2016: A new radiation infrastructure for the Modulat Earth Submodel System (MESSy, based on version 2.51), Geosci. Model Dev. 40, 612-617.

Fuglestvedt, J., Shine, K., Berntsen, T., et al., 2010: Transport impacts on the climate: metrics, Atmos. Environ. 44, 4648-4677.

Hansen, J. et al., 2005: Efficacy of climate forcings, J. Geophys. Res 110, D18104.

Irvine, E., Hoskins, B., Shine, K., 2014: A simple framework for assessing the trade-off between the climate impact of aviation carbon dioxide emissions and contrails for a single flight, Environ. Res. Lett. 9, 064021.

Lee, D.S., et al., 2021: The contribution of global aviation to anthropogenic climate forcing for 2000 to 2018, Atmos. Environ. 244, 117834.

Ponater, M., et al., 2005: On contrail climate sensitivity, Geophys. Res. Lett. 32, L10706.

Ponater, M. et al., 2021: Towards determining the contrail cirrus efficacy, Aerospace 8, 42.

Ramaswamy, V. et al., 2018: Radiative forcing of climate: the historical evolution of the radiative forcing concept, the forcing agents, and their quantifications and applications, Meteor. Monogr. 59, 14-1 – 14.101.

Rap, A., et al., 2010: Estimating the climate impact of linear contrails using the UK Met Office climate model, Geophys. Res. Lett. 37, L20703.

Richardson, T.B., et al., 2019: Efficacy of climate forcings in PDRMIP models, J. Geophys. Res. Atmos. 124, 12824-12844.

Schumann, U., Graf, K., Mannstein, H., Mayer, B., 2012: Contrails: visible aviation induced climate impact, 239-257, In: Atmospheric Physics, Springer, Berlin, Heidelberg.

Teoh, R., et al., 2020: Beyond contrail avoidance: efficacy of flight altitude changes to minimize contrail climate forcing, Aerospace 7, 121.

---

## Author Comment (AC1)

**Response to Reviewer 1**

We would like to thank the anonymous Reviewer for their careful review. We found all the remarks and recommendations to be very relevant and addressing the Reviewer's comments has improved the paper. Below is our response to the comments on a point-by-point basis. The following convention for text fonts is used:

- Review comments in black color
- Our answers in blue color
- *Pieces of text taken from the revised manuscript in red color and in italic font*

**Review**

This is a very interesting, balanced and well thought-out article analysing the use of climate metrics in the context of contrail avoidance. Using the North Atlantic flight tracks from 2019 as a case study, the authors analyse the influence of the choice of climate metric on the decision to re-route around areas of contrail formation. Their conclusion, that the decision to reroute is largely independent of the climate metric, is very interesting and clearly shows that the implementation of contrail avoidance policies need not be unduly hampered by the choice of climate metric. However, as the authors also conclude, since the climate benefit of rerouting depends on the choice of climate metric as well as contrail efficacy and $CO_2$ penalty, further research analysing specific contrail avoidance strategies is well warranted.

Overall, the article is logically presented, well-written and appropriate for ACP. I recommend publication of this article subject to minor revision. I have provided my specific comments and questions below, followed by a few technical remarks and suggestions for improvement.

We thank the Reviewer for their positive assessment of the paper.

**Main comments**

**1)** Choice of background scenario

The authors calculate a total temperature increase for all flights of 16.9 µK in 2039, 13.5 µK in 2069 and 14.0 µK in 2119 (ln. 19). The late increase in temperature between 50 and 100 years after the emission in my opinion requires further description and justification. In Table S1 and Figure 4(a) (third panel), this temperature increase is attributed to $CO_2$.

Following the Reviewer's recommendation, we have discussed this late increase in the discussion of Figure 3: *The 0.5 µK rise between 2069 and 2119 is due to $CO_2$ and the chosen background scenario, SSP4-3.4, which assumes that $CO_2$ concentration decreases after 2080 (Meinshausen et al., 2019). Because the dependence between radiative forcing and $CO_2$ concentration is logarithmic (e.g., Etminan et al., 2016), emitted $CO_2$ has an increased radiative forcing when the atmospheric concentration decreases. In this case, this increase is greater than the decrease from the absorption of $CO_2$ by the ocean and land surfaces, thus leading to a temperature increase from 11.0 µK in 2069 to 12.1 µK in 2119.*

I assume that the emission inventory is treated as a pulse emission in the year 2019? In that case, I can only attribute this temperature increase to a decreasing background $CO_2$ concentration. Looking at Meinshausen et al. (2020), the background $CO_2$ concentration of SSP4-3.4 (ln. 168) does seem to peak in around the year 2080. I thus have a few questions:

**1a)** SSP4-3.4 is characterised by a low $CO_2$ concentration in comparison to the other pathways. Why was SSP4-3.4 chosen for this analysis?

**1b)** I can imagine that the high lingering temperature as a result of $CO_2$ emissions has a significant influence on the weighting of $CO_2$ compared to that of contrails. In particular when more fuel is burned for contrail avoidance. Have the authors considered different background $CO_2$ concentrations?

**1c)** (How) would the results and conclusions presented in this article change if different background $CO_2$ concentrations were to be used? Or in other words, what is the impact/sensitivity of the choice of background $CO_2$ concentration?

We thank the Reviewer for these comments. We address them together.

We chose SSP4-3.4 because it leads to a 2°C warming in the medium term, which currently seems like a plausible trajectory.

The background scenario has indeed a high influence on the perceived climate damage from $CO_2$, for all long-term metrics, such as AGWP100, ATR100 and AGTP100. However, this influence decreases with decreasing time horizon. The Table below shows the dependence of the total climate impact of the flights considered in our study on the assumed scenario, for AGTP20, 50 and 100. The dependence increases with time horizon, with a maximum range for AGTP100, which varies between 8.9 µK for SSP5-8.5 to 14.4 µK for SSP1-1.9. As the Reviewer correctly notes, SSP4-3.4 is characterised by a low $CO_2$ concentration in comparison to the other pathways, so the corresponding AGTP100 value ranks among the highest ones, with a value of 14.0 µK.

| Total warming | AGTP20 | AGTP50 | AGTP100 |
|---|---|---|---|
| SSP1-1.9 | 17.1 µK | 14.2 µK | 14.4 µK |
| SSP1-2.6 | 16.8 µK | 13.6 µK | 13.9 µK |
| SSP2-4.5 | 16.6 µK | 12.7 µK | 12.4 µK |
| SSP3-7.0 | 16.5 µK | 12.1 µK | 10.1 µK |
| SSP4-3.4 | 16.9 µK | 13.5 µK | 14.0 µK |
| SSP4-6.0 | 16.6 µK | 12.5 µK | 11.8 µK |
| SSP5-3.4-OS | 16.3 µK | 12.8 µK | 14.0 µK |
| SSP5-8.5 | 16.4 µK | 11.5 µK | 8.9 µK |

However, as we explained in our study, persistent contrails often induce a warming that is orders of magnitude larger than that due to additional $CO_2$ emissions because of a rerouting, for all $CO_2$-equivalence metrics considered. Therefore, apart from the total warming from all flights as presented above, our qualitative and quantitative conclusions are only weakly dependent on the background scenario used. The Table below shows the number of beneficial reroutings, and the absolute perceived benefit from these reroutings, for AGTP20, 50 and 100 and for different scenarios. For each metric, the number of beneficial reroutings is almost the same, the maximum difference being 4,954 reroutings between SSP5-8.5 and SSP4-3.4 for AGTP100, only 3% of the total number of beneficial reroutings. The perceived absolute benefit also changes only weakly, varying from 1.69 µK for SSP5-8.5 to 1.86 µK for SSP1-1.9.

| Number of beneficial reroutings (absolute benefit in brackets) | AGTP20 | AGTP50 | AGTP100 |
|---|---|---|---|
| SSP1-1.9 | 187,357 (4.62 µK) | 182,359 (2.44 µK) | 178,593 (1.86 µK) |
| SSP1-2.6 | 187,441 (4.58 µK) | 182,595 (2.38 µK) | 178,560 (1.79 µK) |
| SSP2-4.5 | 187,516 (4.58 µK) | 183,282 (2.37 µK) | 179,871 (1.76 µK) |
| SSP3-7.0 | 187,595 (4.57 µK) | 183,860 (2.37 µK) | 182,057 (1.71 µK) |
| SSP4-3.4 | 187,445 (4.61 µK) | 182,755 (2.41 µK) | 178,502 (1.79 µK) |
| SSP4-6.0 | 187,554 (4.59 µK) | 183,494 (2.38 µK) | 180,419 (1.75 µK) |
| SSP5-3.4-OS | 187,653 (4.56 µK) | 183,210 (2.37 µK) | 178,693 (1.82 µK) |
| SSP5-8.5 | 187,626 (4.55 µK) | 184,360 (2.35 µK) | 183,456 (1.69 µK) |

While such results may be interesting to some readers, they would also lengthen the manuscript, hence we did not include a "Sensitivity to the background scenario" section. However, following the Reviewer's comments, we have added some text to address this question: *That background concentration is simulated using the emission scenario SSP4-3.4 in our experiment, which is characterised by a low atmospheric $CO_2$ concentration in comparison to the other pathways. However, conclusions are qualitatively independent on the chosen scenario.*

**2)** Clarification of the methodology

The methodology and analysis used in section 5.2 "Imperfect contrail avoidance" is not clear to me. Is extra $CO_2$ emitted for rerouting, as in the previous sections? How are the number of "imperfect avoidance" flights in Table 1 defined and calculated? How is the rerouting efficiency used to obtain these results?

We renamed the "imperfect contrail avoidance scenario" to "partial avoidance scenario", to clarify the opposition between the partial and the full avoidance scenarios. The partial

contrail avoidance strategy is the same as the full one, but with a decreased rerouting efficiency. That efficiency is calculated following the method described in the paper. The rerouting efficiency is applied to calculate the remaining contrail energy forcing after rerouting, while the additional fuel consumed remains the same. Following the Reviewer's comment, we detailed this methodology in the text: *For each rerouting, contrail energy forcing is scaled by this rerouting efficiency factor, which is less than the 100% efficiency assumed in the full contrail avoidance scenario. [...] For the sake of simplicity, the +1% fuel scenario is used for all reroutings, and results for full and partial avoidance are compared.*

**Minor comments**

ln 35: "Uncertainties [...] and *the* subject of [...]"

Corrected, thanks.

ln 88: "It includes *a* sensitivity analysis [...]"

Corrected.

$\Delta F$ is used to denote "radiative forcing" in the definition of the AGWP in ln. 110, but then also to denote "instantaneous contrail RF" in ln. 137. Since these two definitions are not equivalent, I would suggest to use a different symbol for the calculation of EF_contrail.

Following the Reviewer's comment, we changed $\Delta F$ to $\Delta F_{contrail}$.

In the climate metrics formulae (ln. 110-123), I would recommend performing the integrals from t_0 -> t_0 + H rather than 0 -> H, where t_0 is the emission year (2019). Or, H should be defined more clearly in the text to make the connection between H and t more clear.

Modified as suggested.

The ATR was initially introduced by Dallara et al. (2011), but has subsequently been modified and is generally used as in this article. In the description of the ATR (ln. 118-122), I would therefore make a brief reference to this.

We have added this reference to the definition of ATR, thank you.

ln. 170: The number of simulations used to provide the best estimate and standard deviation, 1726, is very precise. Why was this specific number of simulations used?

To better explain the origin of these 1726 simulations, we modified the corresponding sentence: *We disregard 274 simulations that did not converge to a solution, which can arise with the Monte Carlo approach, and use the remaining 1726 simulations to provide the best estimate and standard deviation.*

Fig 1: I suggest to use the same units in both the caption and the figure (pW and pK)

Modified as suggested.

Fig. 5 and corresponding text: I believe the term "detrimental" is too strong to be a good antonym of "beneficial". It conveys a very strong sense of harm or negative impact, which in my opinion is not appropriate in this context, in particular in light of the uncertainties involved in contrail avoidance. I would advocate for the use of different term here such as "harmful" or equivalent.

While we agree that uncertainties may change results, we think that "detrimental" is a good antonym of "beneficial" in this context. The uncertainties also apply for the "beneficial" term.

ln. 381-382 & ln. 404: "few orders of magnitude more warming" from contrails than from CO2 is ambiguous. How is warming defined in these instances? At what time horizon?

In these instances, the statement is valid for all $CO_2$-equivalence metrics investigated in our work. We specified that point in lines 381-382, but we did not change line 404, as this is stated in the following sentence.

ln. 410-411: "[...] like AGWP20 or ATR20, give a much greater climate benefit [...]" I do not think *give* is the right verb here. I would recommend "suggest" or "calculate".

We changed "give" to "calculate", thanks.

ln. 460: "inflicted by uncertainties" in regards to the ATR is a very loaded wording choice, which I do not think is appropriate. I recommend using more neutral phrasing.

We changed "inflicted with" to "subject to".

---

## Author Comment (AC2)

**Response to Michael Ponater (Reviewer 2)**

We would like to thank Michael Ponater for his careful review. We found all the remarks and recommendations to be very relevant. Based on the review, we have modified our contrail efficacy values and sensitivity analysis, and extended the discussion on different Figures. Below is our response to the comments on a point-by-point basis. The following convention for text fonts is used:

- Review comments in black color
- Our answers in blue color
- *Pieces of text taken from the revised manuscript in red color and in italic font*

**Recommendation**

This is an appropriate and timely contribution to the rapidly emerging discussion on how to assess the climate effect of contrail avoidance measures. It certainly contains a lot of insightful information for scientists and stakeholders alike. The strengths of the paper are to be found in the clearly presented guiding ideas that highlight and illustrate the main problems when quantifying the climate impact gain of contrail avoidance. I am especially pleased with the inclusion of surface temperature change efficacy in the metrics calculations, as this aspect has been disregarded all too often in previous respective assessments. Overall, the conceptual framework and the metrics calculations seem fully adequate to me.

However, I also perceive a number of shortcomings in the presentation of the results, both with respect to a precise description of the methods and to a coherent illustration of technical details. A particular weakness is insufficient explanation of some figures and a lack of hints to the additional material in the supplement. Transferring text from the figure captions and/or from the supplemental material to the main text may be *one* idea for amendment. As the paper deals with a topic of high socio-economic relevance, it is essential to avoid creating any kind of misunderstandings and confusion. Therefore, I encourage the authors to ensure maximum precision all through the text, also when describing the details. I will give a number of major and minor points where I see room for improvement in this respect.

I recommend the paper to be accepted for publication in ACP after those points have been addressed.

We thank the Reviewer for their overall positive assessment of the study. Replies to the perceived shortcomings are below. We have expanded the discussion in places to minimise the possibility of misunderstandings.

**Major Concerns**

1. Formulations in the abstract are generally sound and transparent, except for the final statement that "the choice [of metrics] is ultimately political". I think that could be expressed more clearly, e.g. "the choice of metrics implies to lay a focus on climate impact reduction on shorter or longer time scales, which is ultimately a political decision". In case that is what you meant to say. Any ascertainment should then be naturally emerging from the discussion in the conclusions section.

We changed "the choice is ultimately political" to *"the choice of metric implies a focus on a specific climate objective, which is ultimately a political decision"*. We did not use exact wording proposed by the Reviewer, because metrics are not solely defined by their time scale, but also by their physical definition.

2. The discussion of the results presented in Figure 3 is much to brief and poor guidance is given to the reader to relate the text interpretation to certain figure features. This is emphasized by the fact that the text gives contributions in %, while the figure provides the AGWP/ATR/AGTP values in absolute units. E.g., one feature needing explanation is the $CO_2$ ATR independence from the time horizon, while there is a respective dependency for the end-point metric AGTP. Is there an obvious reason why the relatively large number of contrails with negative EF (line 155) contribute so little to ATR/AGTP? Please realize that these results need to be plausible also to non-experts, and adjust/extend the text accordingly.

We have added additional explanations around Figure 3 in the text, and tried to be as clear as possible. We have explained the percentages with actual numbers from the Figure. We have also added additional statistics in the Methods section to explain why the relatively large number of contrails with negative EF has such a small impact: *The mean energy forcing per unit flown distance of warming contrails is $1.3 \times 10^{11}$ J $km^{-1}$, and is $-2.3 \times 10^{10}$ J $km^{-1}$ for cooling contrails. Although cooling contrails are associated with 20% of contrail-forming flights, their relatively weak energy forcing leads to a total of $2.8 \times 10^{18}$ J removed from the climate system, while warming contrails add $6.6 \times 10^{19}$ J to it, i.e. 23 times more in absolute terms.*

3. In Figure 5, again, the discrepancy between the relative numbers (%) discussed in the text and the absolute numbers (thousands of flights) in the figure makes it difficult to follow the reasoning. I urgently recommend to insert a complementing scale of percentage-numbers along the right y-axis. The whole discussion, however, seems to be rather fuzzy here. The sentence "The number of "lower risk" reroutings is 550% larger" (hardly comprehensible in itself) seems to be the only interpretation of the second row of Figure 5, which is clearly insufficient to convince the reader of anything. Please, extend the discussion and refer to exactly the columns you are interpreting.

4.

As the Reviewer suggested, we have added a new scale showing percentages along the y-axis on the right-hand side. The discussion has been extended, and the sentence the Reviewer refers to has been modified to be more comprehensible: *The number of "lower risk" reroutings is 5.5 times larger when no additional fuel is emitted than in the +1% fuel scenario, because the condition on maximum additional fuel is always met.*

5. Some confusion already exists in the (sparse) literature on the choice of numbers for certain efficacy parameters in the "radiative forcing" (RF) framework and in the "effective radiative forcing" (ERF) framework, and I feel that your paper could help to amend respective inconsistencies. The paper correctly distinguishes between a contrail "total efficacy" (line 180), which results as the product of an ERF/RF factor (originating from rapid radiative adjustments induced by the contrail forcing) and the ERF related efficacy (line 178) that originates from surface temperature induced

radiative feedbacks. You have chosen a value of 0.35 for the total efficacy by multiplying the ERF/RF factor (0.35) from Bickel et al. (2020) with a value of 1 for the ERF related efficacy "for lack of a better estimate". Yet, there is an estimate of the ERF related efficacy and it's given in line 377! While there may be a good reason to choose the values as you do, any source of misunderstanding in the context should be eliminated.

When it comes to uncertainty considerations in section 5.3, the discussion starts with the sentence "Contrail efficacy has been set so far to the best estimate of 0.35 from Bickel et al. (2020)." This wording is confusing, because the Bickel et al.'s value is actually an ERF/RF factor – as you correctly introduce in section 3. You probably meant to repeat the section 3 definition in a nutshell, but actually this combination of ERF/RF factors and efficacies emphasizes a respective simplifying approach by Lee et al. (2021), who calculated a best estimate for the contrail cirrus ERF/RF factor (value 0.42 – as referenced in line 222) from three individual model results of which two (Ponater et al., 2005, and Rap et al., 2010) had rather provided a total efficacy value. It should be easy to avoid such simplifications in the context of your paper.

An analogous criticism applies to your choice of "contrail efficacy" values 0.23 and 0.51 (line 377) referring to the confidence interval given by Bickel et al. (2020) – but again that was for their ERF/RF factor. For choice of a lower bound for contrail total efficacy, the ERF related efficacy best estimate from Bickel (2023), i.e., 0.38, is combined with the lower estimate of the ERF/RF factor from Bickel et al. (2020). This is largely okay in a qualitative sense, but it neglects that Bickel (2023) also provides a total efficacy value in his Table 4.3., i.e. 0.21, by combining their 0.38 value for ERF related efficacy with the ERF/RF factor within their own model framework – which is more consistent. Still, given that Bickel's (2023) total efficacy value (0.21) will also have an uncertainty (which is not quantified in his thesis), your estimated parameter range for total efficacy appears quite reasonable to me.

We thank the Reviewer for this very important comment. As the Reviewer correctly points out, the paper was not clear enough about contrail efficacy, sometimes mixing ERF-to-RF ratio with total contrail efficacy. Moreover, we did not consider other estimates of total contrail efficacy than that from Bickel (2023). To better represent current knowledge and for an increased readability, we changed the value we use for total contrail efficacy. Instead of using 0.35, which was the product of the best estimate for the ERF/RF ratio from Bickel et al. (2021) 0.35, and 1 for the "Ponater" efficacy, we now use the average value between the three values available in the literature as given by the Reviewer: 0.59 (Ponater et al., 2005), 0.31 (Rap et al., 2010) and 0.21 (Bickel, 2023). This leads to a total contrail efficacy of 0.37. All the Figures and numbers in the manuscript have been changed accordingly, but the conclusions remain the same, as the new value is close to the previous value of 0.35. The description of total contrail efficacy remains the same, but the sentence about the chosen contrail efficacy has been changed: *It is set to 0.37 in this study, which is the average of the three available estimates in the literature, specifically 0.59 (Ponater et al., 2005), 0.31 (Rap et al., 2010) and 0.21 (Bickel, 2023).*

Following the Reviewer's comment, we have also changed the values used in the sensitivity analysis of Section 5.3. Instead of using 0.23 and 0.51 which were taken from Bickel et al. (2021) and Bickel (2023), we now use the three best estimates of total contrail efficacy: 0.21,

0.31 and 0.59. To continue considering more extreme values of contrail efficacy, we retain the 0.05, 0.1 and 1 values.

6. The total efficacy as chosen in the paper (0.35) "is assumed to apply for all contrails" (line 182). This may well be acceptable for the conceptual targets of this paper, yet it implies that the rapid radiative adjustments (controlling ERF/RF), as well as the surface temperature induced radiative feedbacks, are independent of where and when a contrail develops. This is, obviously, a very unlikely assumption. The authors appear to be well aware of this fact, as (line 389) they write that the climate benefit from contrail avoidance, when calculated accordingly, is only valid "when considering rerouted flight together, not on a flight-to-flight basis". This overall assessment is likely correct (or, at least, tenable), but I feel that an extra warning is necessary here, in order to avoid the impression that RF (or the energy forcing, EF) of an individual contrail (or flight) can safely be converted to ERF by using a uniform factor derived from global considerations.

This is indeed an important assumption. Following the Reviewer's comment, we have added an extra caveat when defining the contrail efficacy : *The same factor is assumed here to apply to all contrails, but it is likely that contrail efficacy varies depending on the location and time of a flight. Nevertheless, this assumption is adopted here in the context of this study, where results are considered an average over a population of North Atlantic flights.*

7. There is a further potential source of error when introducing efficacy parameters in metrics calculations, as the CoCiP model primarily uses energy forcing (EF) to assess the climate impact of individual flights (Schumann et al., 2012; Teoh et al., 2020), which can then be used to calculate a global instantaneous radiative forcing (RFinst). However, global ERF/RF, climate sensitivity and efficacy parameters (e.g. Bickel et al., 2020; Lee et al., 2021; Bickel, 2023) are usually given on the basis of the stratosphere adjusted radiative forcing (RFadj). Switching between RFadj and RFinst makes not much of a difference for contrails (Dietmüller et al., 2016, their Table 2), but strongly matters when calculating the tradeoff with the radiative effect induced by additional CO2 emissions. I presume that, concerning your paper, OSCAR uses additional fuel consumption (rather than a CO2 EF provided by CoCiP) to calculate the CO2 RFadj. I deem it important that the text should not leave open any doubt on this (line 160), as otherwise a combination with the published global ERF/RF factors (and efficacies) would be inconsistent (see efficacy fluctuations for various RF definitions, as reported by Richardson et al., 2019, their Table 1).

Following the Reviewer's comment, we have explicitly mentioned that OSCAR computes the time evolution of adjusted radiative forcing from the provided flight $CO_2$ emissions. Additionally, we clarified that OSCAR, in its version 3.1.1 that we used, uses the RF $CO_2$ formula from Myhre et al. (2013).

**Specific remarks/recommendations**
Pg. 1, l. 17: "… defined here as nine combinations of different definitions …" sounds somewhat strange. Perhaps "… represented here as nine combinations …"?
Modified as suggested.

Pg. 1, l. 19: I recommend to write "Under an idealized scenario where …" to avoid any misunderstanding.

Modified as suggested.

Pg. 2, l. 33: As there is no extra definition section of radiative forcing in this paper, I recommend to insert another sentence here (before "According to"): "The climate effect of CO2 and non-CO2 contributions has usually been compared in terms of several radiative forcing parameters (e.g., Ramaswamy et al., 2018) or dedicated metrics (e.g., Fuglestvedt et al., 2010).

Following the Reviewer's comment, we added the sentence: *"The $CO_2$ and non-$CO_2$ climate effects have usually been compared in terms of radiative forcing (e.g., Ramaswamy et al., 2018) or other dedicated metrics (e.g., Fuglestvedt et al., 2010)."* We did not use exact wording proposed by the Reviewer, because we think that "radiative forcing" is clearer than "several radiative forcing parameters".

Pg. 4, l. 103: "It is impractical …", this statement demands a rationale or a reference.

Following the Reviewer's comment, we added an explanation for our statement: *[...] because the climate impact of such a small emission pulse would be lost in the internal variability of the Earth system, requiring too many climate model simulations to identify it robustly.*

Pg. 6, l. 170: "… 1726 simulations …"; Why is this selection necessary? Is this relevant for the outcome?

In response to a comment by the other Reviewer, and to better explain the origin of these 1726 simulations, we modified the corresponding sentence: *We disregard 274 simulations that did not converge to a solution, which can arise with the Monte Carlo approach, and use the remaining 1726 simulations to provide the best estimate and standard deviation.*

Pg. 6, l. 175: A personal remark at this point: You choose to label the ERF related efficacy (line 178) that originates from surface temperature induced radiative feedbacks as the "Ponater efficacy". While I feel flattered by this labelling, the term might still be misleading, as the cited reference by Ponater et al. (2021) deals with both the RF and the ERF related efficacy parameter and their interrelation, but without giving numbers. Different definitions of efficacy were introduced by Hansen et al. (2005), but the first dedicated calculation of the ERF related efficacy for contrails was given by Bickel (2023), yielding a value of 0.38, as is mentioned in your paper but only later (line 377).

Following this comment and the general recommendations of the Reviewer for increased clarity, we changed the name "Ponater efficacy" for "ERF-based efficacy". Additionally, we have added the Bickel (2023) reference to the initial definition, line 178.

Pg. 6, l. 180: Please, add references to Hansen et al. (2005) and Richardson et al. (2019).

Added as suggested.

Pg. 6, l. 182: "… the same factor is assumed for all contrails." Does this mean that each contrail EF is multiplied with the assumed contrail efficacy? Is this relevant for OSCAR or does that model only receive global mean contrail RF as an input? – See major concerns.

The contrail RF is indeed multiplied with the contrail efficacy. We have clarified this point in the text. This is relevant because OSCAR does not differentiate between RF from contrails or any other source of RF, so the contrail-specific efficacy is imposed before the RF time series is provided to OSCAR.

Pg. 6, l. 187: Recommended modification: "… are illustrated in an idealized but representative way by Figure 1, …"
Modified as suggested.

Pg. 7, l. 202-204: This sounds as if the ocean surface layer absorbs more or less instantaneously the whole energy provided by any forcing and then returns the temperature signal slowly back to the atmosphere, but such a notion would be inconsistent with the known large ocean heat capacity. However, I presume that's not what you mean. Please make it plain that the time limit for the heat transfer from the atmosphere to the ocean is prescribed in your scenario through the length of the "pulse" for which the contrail forcing lasts.
We have modified the sentence, so that it is clear that the ocean absorbs the excess energy over time: *"[...] the ocean absorbs the energy perturbation resulting from the radiative forcing **over the time the forcing is exerted**, increasing its heat content [...]"*

Pg. 8, l. 229: Lee et al. (2021), Appendix E, explicitly states: "When calculating the contrail cirrus ERF, the error range given refers to the error range of contrail cirrus RF and not ERF." Hence, I recommend to modify your formulation to "… confidence level in contrail cirrus ERF, as given by Lee et al. (2021) on the basis of corresponding RF uncertainty considerations."
Modified as suggested.

Pg. 9, l. 244: Choice of the headlines in this section 5 seems a bit incoherent to me. One idea could be to title the subsection here as "5.1 Assessment of an idealized contrail avoidance scenario"; then replace the heading of what is now sub-section "5.1 Sensitivity to additional CO2" by "5.2 Sensitivity to the amount of additional CO2 emissions"; and then replace the heading of what is now "5.2 Imperfect contrail avoidance" by "5.3 Sensitivity to imperfect contrail avoidance". The 5.3 heading could remain as it is (but then as 5.4). Please, consider this suggestion as a potential improvement.
Modified as suggested.

Pg. 11, l. 290: "Figure 5 shows …", seems bad wording to me as Figure 5 is in fact dealt with only in the following sub-section. "As will be shown in the next sub-section" may be an improvement. But see major concerns (above) with respect to Figure 5.
Modified as suggested.

Pg. 12, l. 305: "…into the climate damage category, as defined in the previous sub-section"
Modified as suggested.

Pg. 12, l. 310: "550% larger"; I fail to relate this statement to any feature in Figure 5, nor do I understand what the statement actually means (larger than what?). But see major concerns above.
We address this comment in the major concern #3 of the Reviewer.

Pg. 14, l. 335: "There is no simple relationship between addition CO2 and contrail EF": True indeed! The statement could be moved to the begin of the sub-section to motivate why "The results presented in Figure 5 are idealised" (line 332), despite the fact that the additional fuel increment is varied in that figure.

While we agree that this statement applies to our entire study, in particular to the results of Figure 5, we think it has more strength when assessing contrail avoidance strategies. We therefore chose to keep this statement at its current location.

Pg. 14, l. 342: "… the rerouting fails to reduce the energy forcing …" Completely? Why? I fail to perceive the logic in this assumption.

This is only a limit case where a flight would produce originally a persistent contrail along its entire distance. In such a case, we assume that the ISSR is too large to be avoided. This is of course a simplification of a more complex reality, but the statistics on the rerouting efficiency (the 5th percentile is 31%) shows that this limit case, where rerouting efficiency would be 0%, only represents a small number of cases.

Pg. 14, l 344: "… the average rerouting efficiency is 0.71 …"; How is this number calculated? And how is it, then, used to modify the original results presented in Figure 4? Is this addressed in the following paragraph?

This number is calculated using the dataset we used in the previous subsections. To clarify, we modified the sentence to: *For the contrail-forming flights flown over the North Atlantic in 2019, we calculate that the average rerouting efficiency for all reroutings would be 71%.* We also added a sentence above, to explain how the rerouting efficiency is used: *For each rerouting, contrail energy forcing is scaled by this rerouting efficiency factor, which is less than the 100% efficiency assumed in the full contrail avoidance scenario.*

p. 14, l. 351: "In contrast, … temperature rise in 100 years … by only 3%". Which line in Table 1 are you referring to here, which numbers are you comparing? Is "warming in 100 years" (l. 342) referring to something different than "temperature rise in 100 years" (l. 352)? Please, be precise and refer to the actual metric you are addressing.

We have made it clear that we use AGTP100, and added the actual numbers from Table 1.

p. 14, l. 356: "… the absolute benefit comes from flights for which the impact of the additional emitted CO2 is much greater than the impact of the avoided contrail"; the statement strikes me as illogical. What do you mean by this?

We thank the Reviewer for this comment; this is indeed a mistake, which has been corrected to: *"… the absolute benefit comes from flights for which the impact of the additional emitted CO2 is much **lower** than the impact of the avoided contrail"*

p. 15, l. 371: "Contrail efficacy has been set to the best estimate of 0.35 from Bickel et al. (2020). This statement is confusing as Bickel et al. provide an ERF/RF factor. Please, consider the efficacy related part of the major concerns above.

We address this comment in the major concern #4 of the Reviewer.

p. 16, l. 396: In some contrast to the results section, formulations in the conclusions section are generally sound and precise.

We thank the Reviewer for this comment.

p. 16, l. 401: Some more references to previous work would be fine here. Immediately to my mind come Deuber et al. (2013) and Irvine et al. (2014).
We have added these two references.

**References:**

Bickel, M., et al., 2020: Estimating the effective radiative forcing of contrail cirrus, J. Clim. 33, 1991-2005.

Bickel, M., 2023: Climate impact of contrail cirrus, DLR-Forschungsbericht 2023-14, PhD Thesis, Ludwig-Maximilians-Universität München, https://edoc.ub.uni-muenchen.de/view/autoren/Bickel=3AMarius=3A=3A.html

Deuber, O., Matthes, S., Sausen, R., et al., 2013: A physical metric-based framework for evaluating the climate trade-off between CO2 and contrails – the case of lowering aircraft flight trajectories, Environ. Sci. Policy 25, 176-185.

Dietmüller, S. et al., 2016: A new radiation infrastructure for the Modulat Earth Submodel System (MESSy, based on version 2.51), Geosci. Model Dev. 40, 612-617.

Fuglestvedt, J., Shine, K., Berntsen, T., et al., 2010: Transport impacts on the climate: metrics, Atmos. Environ. 44, 4648-4677.

Hansen, J. et al., 2005: Efficacy of climate forcings, J. Geophys. Res 110, D18104.

Irvine, E., Hoskins, B., Shine, K., 2014: A simple framework for assessing the trade-off between the climate impact of aviation carbon dioxide emissions and contrails for a single flight, Environ. Res. Lett. 9, 064021.

Lee, D.S., et al., 2021: The contribution of global aviation to anthropogenic climate forcing for 2000 to 2018, Atmos. Environ. 244, 117834.

Ponater, M., et al., 2005: On contrail climate sensitivity, Geophys. Res. Lett. 32, L10706.

Ponater, M. et al., 2021: Towards determining the contrail cirrus efficacy, Aerospace 8, 42.

Ramaswamy, V. et al., 2018: Radiative forcing of climate: the historical evolution of the radiative forcing concept, the forcing agents, and their quantifications and applications, Meteor. Monogr. 59, 14-1 – 14.101.

Rap, A., et al., 2010: Estimating the climate impact of linear contrails using the UK Met Office climate model, Geophys. Res. Lett. 37, L20703.

Richardson, T.B., et al., 2019: Efficacy of climate forcings in PDRMIP models, J. Geophys. Res. Atmos. 124, 12824-12844.

Schumann, U., Graf, K., Mannstein, H., Mayer, B., 2012: Contrails: visible aviation induced climate impact, 239-257, In: Atmospheric Physics, Springer, Berlin, Heidelberg.

Teoh, R., et al., 2020: Beyond contrail avoidance: efficacy of flight altitude changes to minimize contrail climate forcing, Aerospace 7, 121.

---

## Referee Report (RR1)

2nd review of the manuscript "The importance of an informed choice of $CO_2$-equivalence metrics for contrail avoidance" by A. Borella et al. (egusphere-2024-347)

**Recommendation**

The authors have addressed my comments in a careful and very satisfactory way, and consequently I now recommend their paper for publication in Atmospheric Chemistry and Physics.

I still give some further suggestions below for consideration by the authors, partly on text I left uncommented in the first review (sorry for this!). However, my recommendation "accept" in no way depends on whether or not these additional suggestions will be accounted for.

**Optional suggestions**

l. 91: As the term "efficacy" is itself somewhat un-specific but has a dedicated meaning in the framework of this paper, it might be sensible to introduce it as "the efficacy of contrail radiative forcing to induce surface temperature changes" (replacing "the contrail efficacy").

l. 117: "… it is not an explicit measure of the climate response." I now feel that this statement calls for another reference, which could be Fuglestvedt et al. (2010) in view of their respective considerations at lines 4655, 4656.

l. 174: "adjusted radiative forcing": since the term is not clearly defined within the paper, I suggest to write "stratospheric-adjusted radiative forcing", and to add references to Hansen et al., 2005, and to Forster et al., 2007 (AR4, chapter 2).

l. 177: Suggestion (after "… considerations"): "Note that use of instantaneous contrail RF (corresponding to EF) for contrails and use of stratospheric-adjusted RF for $CO_2$ is not inconsistent, as instantaneous and stratospheric-adjusted RF do not differ significantly for contrails (Dietmüller et al., 2016)".

l. 201: After "AGWP" you might consider to add another explicit reference: "Fuglestvedt et al. (2003, their Eq. 7)".

l. 243: "The distribution", it might be specified which distribution is addressed.

l. 349: "The number of …", The meaning of this sentence is still somewhat cryptical to me, especially with respect to the "condition of maximum additional fuel". By the way "5 times larger" or "5.5 times larges" (as you write in your reply)?

l. 362: "low energy contrails", perhaps improve to "low EF contrails"?

l. 475: Is there a reference to back the statement made in preceding sentence?

---

## Author Response (AR2)

We thank the Reviewer for their positive assessment of the revised manuscript, and for their new comments. We address them point by point below. The review is in black, our answers in blue, and modified text in the manuscript in italic red.

I thank the authors for their work and detailed replies to my comments. All of my previous comments were sufficiently addressed. I especially appreciate the detailed response to the dependence of the results on the background emissions scenario. This is interesting work which in my opinion could warrant a description in the supplement, but I leave that decision to the authors.

Overall, I commend the authors on their interesting work and recommend publication of this article subject to one further minor revision, which I have outlined below.

**Minor revision**

In Section 5.4 (previous Section 5.3), the authors vary the contrail efficacy. In the preprint, a best estimate of 0.35 was used, which was now modified to 0.37. The authors used this best estimate for normalisation in Figure 6. However, it seems that there may have been a mistake in the normalisation or visualisation of the results, since in the new Figure 6 unity is shown for an efficacy of 0.31. Could the authors please check their (visualisation) code to ensure that the correct best estimate is used for all results?
This is indeed an error, thank you for spotting it. It is corrected.

**Technical comments/suggestions**

ln. 82: "time horizon" rather than just "horizon"
Added as suggested.

ln. 116-7: "AGWP is a time-integrated metric, and because it is based on radiative forcing, it is not an explicit measure of the climate response." Does this final clause refer to the temporal integration or to the radiative forcing? What is meant by an "explicit measure"?
This final clause relates to the radiative forcing. This is not an explicit measure of climate response, because radiative forcing is only an intermediate quantity between the perturbation and the climate response. We added a reference to Fuglestvedt et al. (2010) to support this statement, as suggested by Reviewer 2.

ln. 120: In line with the other definitions, I would suggest $\Delta T(t\_0 + H)$ here as well
Added as suggested.

ln. 390: I thank the authors for including an extra sentence here describing the rerouting efficiency factor. Since a rerouting efficiency factor of 100% corresponds to the case that all contrails are avoided, I suggest the following modification: "For each rerouting, contrail energy forcing is *inversely* scaled by this rerouting efficiency factor" (or words to this effect). I believe this makes the definition of the factor more clear.
Modified as suggested.

2nd review of the manuscript "The importance of an informed choice of CO2-equivalence metrics for contrail avoidance" by A. Borella et al. (egusphere-2024-347)

We thank Michael Ponater for his positive assessment of the revised manuscript, and for his new comments. We address them point by point below. The review is in black, our answers in blue, and modified text in the manuscript in italic red.

**Recommendation**

The authors have addressed my comments in a careful and very satisfactory way, and consequently I now recommend their paper for publication in Atmospheric Chemistry and Physics.

I still give some further suggestions below for consideration by the authors, partly on text I left uncommented in the first review (sorry for this!). However, my recommendation "accept" in no way depends on whether or not these additional suggestions will be accounted for.

**Optional suggestions**

l. 91: As the term "efficacy" is itself somewhat un-specific but has a dedicated meaning in the framework of this paper, it might be sensible to introduce it as "the efficacy of contrail radiative forcing to induce surface temperature changes" (replacing "the contrail efficacy").
Modified as suggested.

l. 117: "… it is not an explicit measure of the climate response." I now feel that this statement calls for another reference, which could be Fuglestvedt et al. (2010) in view of their respective considerations at lines 4655, 4656.
Reference added as suggested.

l. 174: "adjusted radiative forcing": since the term is not clearly defined within the paper, I suggest to write "stratospheric-adjusted radiative forcing", and to add references to Hansen et al., 2005, and to Forster et al., 2007 (AR4, chapter 2).
Modified as suggested.

l. 177: Suggestion (after "… considerations"): "Note that use of instantaneous contrail RF (corresponding to EF) for contrails and use of stratospheric-adjusted RF for CO2 is not inconsistent, as instantaneous and stratospheric-adjusted RF do not differ significantly for contrails (Dietmüller et al., 2016)".
Added as suggested, with small modifications with no impact on the meaning: *The use of instantaneous RF (corresponding to EF) for contrails and the use of stratospheric-adjusted RF for $CO_2$ is not inconsistent, as instantaneous and stratospheric-adjusted RF do not differ significantly for contrails (Dietmüller et al., 2016).*

l. 201: After "AGWP" you might consider to add another explicit reference: "Fuglestvedt et al. (2003, their Eq. 7)".
Reference added as suggested.

l. 243: "The distribution", it might be specified which distribution is addressed.
We added "The distribution of contrail energy forcing per flown distance" as suggested.

l. 349: "The number of …", The meaning of this sentence is still somewhat cryptical to me, especially with respect to the "condition of maximum additional fuel". By the way "5 times larger" or "5.5 times larges" (as you write in your reply)?

This is indeed 5 times larger (our reply was erroneous). We also modified the sentence to increase its clarity; hopefully it is now no more cryptic: *The number of "lower risk" rerouting is 5 times larger when no additional fuel is emitted compared to the +1% fuel scenario. This is because our definition of "lower risk" rerouting relies on a maximum amount of additional fuel, and this condition is always met when no additional fuel is emitted.*

l. 362: "low energy contrails", perhaps improve to "low EF contrails"?

As we make clear in the second part of the sentence that this relates to EF ("with EF per flown kilometre from …"), we prefer keeping the sentence as is.

l. 475: Is there a reference to back the statement made in preceding sentence?

We added the following reference: Dalmau Codina, R.; Melgosa Farrés, M.; Vilardaga Garcia-Cascón, S.; Prats Menéndez, X. A fast and flexible aircraft trajectory predictor and optimiser for ATM research applications. In Proceedings of the International Conference on Research in Air Transportation, Catalonia, Spain, 25–29 June 2018.